

# A novel semi-direct method to measure OH reactivity by chemical ionisation mass spectrometry (CIMS)

Jennifer B. A. Muller, Thomas Elste, Christian Plass-Dülmer, Georg Stange, Robert Holla, Anja Claude, Jennifer Englert, Stefan Gilge, Dagmar Kubistin

Meteorologisches Observatorium Hohenpeissenberg, Deutscher Wetterdienst (DWD), 82383 Hohenpeissenberg, Germany

*Correspondence to*: Dagmar Kubistin (Dagmar.Kubistin@dwd.de)

**Abstract.** An operational chemical ionisation mass spectrometer (CIMS) for hydroxyl radical (OH) and sulfuric acid ($H_2SO_4$) concentration measurements was adapted to include observations of OH reactivity, which is the inverse of OH lifetime, for long-term monitoring at the Global Atmosphere Watch (GAW) site Hohenpeissenberg (MOHp), Germany. OH

measurement using CIMS is achieved by reacting OH with $SO_2$, leading to the production of $H_2SO_4$ which is then detected. The adaptation for OH reactivity consists of the implementation of a second $SO_2$ injection, at a fixed point further down flow in the sample tube to detect the OH decay caused by reactions with OH reactants present in the sample.

The method can measure OH reactivity between below 1 and 40 $s^{-1}$ with the upper limit due to the fixed positioning of the second $SO_2$ injection. To determine OH reactivity from OH concentration measurements, the reaction time between the two

titration zones and OH wall losses in the sample tube need to be determined accurately through OH reactivity calibration. Potential measurement artefacts as a result of HOx recycling in the presence of NO have to be considered. Therefore NO contamination from gases used in instrument operation must be minimised and ambient NO must be measured concurrently to determine the measurement error.

This CIMS system is shown here to perform very well for OH reactivity below 15 $s^{-1}$ and NO concentrations below 4 ppb,

both values that are rarely exceeded at the MOHp site. Thus when deployed in suitable chemical environments, this method can provide valuable continuous long-term measurements of OH reactivity. The characterisation utilises results from chamber, laboratory and modelling studies and includes the discussion and quantification of sources of uncertainties.

## 1 Introduction

Trace gas removal in the troposphere is largely controlled by the hydroxyl radical (OH). In a time when the chemical

composition of the atmosphere is significantly and rapidly changing (Monks et al., 2009) we need to better understand what processes determine the oxidative self-cleaning capacity of the atmosphere as well as monitor how this oxidation efficiency is evolving over longer timescales. The estimation and measurement of OH reactivity, termed also total OH loss rate ($s^{-1}$), contributes to elucidating oxidant budgets and thus our understanding of atmospheric photochemical cycling and oxidation capacity (e.g. Stone et al., 2012; Rohrer et al., 2014; Yang et al., 2016). As envisioned in a recent "roadmap for OH





reactivity", continuous OH reactivity measurements at long-term monitoring stations should become permanent additions to the standard observation portfolio in time (Williams and Brune, 2015). The purpose of this paper is to both present a new measurement method for OH reactivity and demonstrate the suitability of this approach for long-term measurements at the Global Atmosphere Watch (GAW) monitoring station Hohenpeissenberg (WMO, 2006) and other similar stations.

OH reactivity can be measured using a range of different approaches. Established techniques have recently been summarised by e.g. Hansen et al. (2015), Yang et al. (2016) and Fuchs et al. (2017). In principle OH reactivity can either be measured by directly probing the pseudo first order OH decay curve occurring in the presence of OH reactants or by observing the relative change in OH reactivity of a chosen OH reactant not present in the atmosphere. In the direct approach, OH is detected by Laser Induced Fluorescence (LIF) (Kovacs and Brune, 2001) or Chemical Ionisation Mass Spectrometry (CIMS) (McGrath,

2010). The more indirect method is called Comparative Reactivity Method (CRM) and here Proton-Transfer-Reaction Mass Spectrometry (PTR-MS) (Sinha et al., 2008) or a Gas Chromatographic Photo-Ionisation Detector (GC-PID) (Nölscher et al., 2012) is used for detection of the commonly chosen OH reactant pyrrole. For the direct method measuring the decay of OH concentration, a flow tube set up forms an integral part of the measurement system. Either the flow tube is used with a laser-photolysis pump and probe approach (Sadanaga et al., 2004) or it is used with a moveable injector (Kovacs and Brune, 2001;

Ingham et al., 2009; Hansen et al., 2015). The combination of a moveable injector inside a flow tube combined with OH detection by CIMS is described by McGrath (2010) and the method described here uses a CIMS for OH detection, but with a novel approach of measuring the OH decay.

The development and design of the method to measure OH reactivity with CIMS as described here arose out of an existing and ongoing long-term measurement setup for OH and $H_2SO_4$ at the Meteorological Observatory Hohenpeissenberg

(MOHp) in Germany (Berresheim et al., 2000; Rohrer and Berresheim, 2006). The measurement of OH reactivity was added to the continuous OH and $H_2SO_4$ measurement routine in 2009. The method was conceived with the following requirements, which were all achieved: maintaining OH and $H_2SO_4$ observational continuity, maintaining stability and appropriate time resolution for all measurements, a design optimised for expected OH reactivity up to about 20 s$^{-1}$ with high level of automation and infrequent need for manual calibration. No moving parts, as would be the case of a moveable injector, had to

be installed, producing a suitable robustness for long-term operation.

In this paper, we first describe the measurement principles of OH reactivity by CIMS as well as the model set up used to investigate instrument internal chemical processes in Section 2. Section 3 deals with the characterisation of OH reactivity measurements, including the quantification of errors from OH reactivity calibration and instrument internal HOx recycling in the presence of NO. Approaches for minimizing uncertainties and correcting systematic errors are suggested. The quality of

OH reactivity measurements are examined in Section 4, where CIMS measurements are compared to those from a LIF measurement system during a chamber study (Fuchs et al., 2017). Continuous OH reactivity measurements of ambient air at the MOHp during the year 2015 are also presented in Section 4, before concluding remarks in final Section 5.



## 2 Measurement principles using CIMS

### 2.1 OH measurement

The method for measuring OH reactivity using CIMS builds on OH concentration measurements that have been described extensively elsewhere (in principle e.g. Eisele and Tanner, 1991; Tanner et al. 1997; and specifically for this CIMS instrument and set up in Berresheim et al., 2000; Schlosser et al., 2009).

Briefly, OH radicals are converted to sulfuric acid molecules ($H_2SO_4$) by addition of $SO_2$. $H_2SO_4$ is then ionised by charge transfer utilising negative chemical ionisation with $NO_3^-$ reactant ions. The $NO_3^-$ reactant ions are produced by flowing synthetic air enriched with $HNO_3$ over a radioactive alpha emitter ($^{241}$Am) before entering the ionisation region as sheath flow at atmospheric pressure. Both $NO_3^-$ reactant and $HSO_4^-$ product ions are then monitored by quadrupole mass spectrometry to determine concentrations (Fig. 1).

The CIMS system is frequently calibrated to obtain absolute OH concentrations and to monitor changes to the calibration factor which is indicative of stability in instrument sensitivity. For this, a special calibration unit forms an integral part of the measurement set up (Tanner et al., 1997) and is described in detail in Berresheim et al. (2000). Ambient water vapour is photolysed at 185 nm produced by a UV Pen-Ray lamp to produce a known OH concentration in front of the sample tube, with about 5 ms travel time before $SO_2$ is injected. The Hg lamp also produces a 253 nm emission line which is suppressed by an interference filter (CaF2, 185 nm, Laseroptik) to avoid OH production by $O_3$ photolysis and the reaction of the produced O($^1$D) with water vapour (Fig. 1). The photolysis of water vapour produces an equal amount of OH and H radicals (R1), the latter reacting rapidly with $O_2$ to produce the hydroperoxy radical $HO_2$ (R2).

$H_2O + h\nu$ ($\lambda$ = 185 nm) → H + OH       (R1)

$H + O_2 + M$ → $HO_2 + M$       (R2)

OH is then measured indirectly through the conversion of OH to $H_2SO_4$ which occurs in the flow tube chemical reactor in front of the ionisation region. The OH titration proceeds via reactions (R3-R5) within approximately 20 ms, after the addition of $SO_2$ to the sample tube flow via front injectors F1 or F2 achieving a $SO_2$ concentration of about 10 ppm in the sample tube chemical reactor (Fig. 1).

$SO_2 + OH + M$ → $HSO_3 + M$       (R3)

$HSO_3 + O_2$ → $SO_3 + HO_2$       (R4)

$SO_3 + H_2O + M$ → $H_2SO_4 + M$       (R5)

With 13 SLM sample flow through the 0.019 m diameter chemical reactor sample tube, the total residence time within the flow tube is circa 0.9 s. This is long enough for HOx recycling to become significant inside the flow tube and the





measurement artefacts from recycling thus need to be considered and accounted for. When nitric oxide (NO) is present, HOx recycling and thus OH production within the sample tube can occur via reactions (R6-R8):

NO + HO$_2$ → NO$_2$ + OH  (R6)

NO + RO$_2$ → NO$_2$ + RO  (R7)

RO + O$_2$ → HO$_2$ + product  (R8)

Other important recycling reactions, such as HO$_2$-radical reactions including those with halogens (e.g. Cl) as well as the reaction of HO$_2$ and O$_3$, are considered negligible here because of the continental measurement location and time frames

relevant for the CIMS system. In order to minimise the effect of NO induced HOx recycling, excess propane (C$_3$H$_8$) is added via the rear injectors (R1/R2 in Fig. 1) after the OH titration with SO$_2$ has completed. Propane concentration in the sample is then about 315 ppm, leading to OH loss by reaction with propane 20 times faster than with SO$_2$, and about 97 % OH is consumed by propane.

An additional measurement cycle is performed to quantify the residual recycled OH, ambient H$_2$SO$_4$ and any other species

such as stabilised Criegee intermediates (e.g. Mauldin et al., 2012; Novelli et al., 2014) that react with SO$_2$ to form a H$_2$SO$_4$ measurement artefact. After the titration measurement (or "signal" mode, i.e. SO$_2$ injection at front and propane injection at rear), a "background" measurement is made. This involves adding both SO$_2$ and propane at the front injector, leading to the quantification of the contribution from recycling and other artefacts to the sulfuric acid concentration. The final OH concentration is thus the difference between signal and background measurements.

For the OH reactivity measurement in ambient air, the ultimate final OH concentration measurement for a given titration zone (Fig. 1), i.e. [OH]$_{T1}$ and [OH]$_{T2}$, is the final OH concentration without UV light subtracted from the final OH concentration with the UV lamp on. This way any contribution from ambient OH on the OH reactivity measurement, typically at the order of a few percent, is removed.

## 2.2 OH reactivity (kOH) measurement

The total loss rate of OH (kOH), or OH reactivity, is defined as the inverse of OH lifetime and described by the pseudo first order decay of OH in the presence of OH reactants (Kovacs and Brune, 2001):

$$[OH](t) = [OH]_0 \times e^{kOH \times t} \qquad (1)$$

with $kOH = \sum k_{Xi+OH} \times [X_i]$ and $X_i$ for the OH reactants.

Whilst some measurement techniques effectively capture the full OH decay curve (Sadanaga et al., 2004), the CIMS technique is based on the lawful logarithmic relation of any two points of OH concentration at time t$_1$ and time t$_2$ (i.e. [OH]$_{t1}$





and $[OH]_{t2}$) on the decay curve, from which OH reactivity can be calculated if the (reaction) time $\Delta t = (t_2 - t_1)$ between the two points and the CIMS system specific OH wall loss rate $k_w$ are known.

$$\frac{-d[OH]}{dt} = (kOH + k_w) \times [OH] \qquad (2)$$

$$[OH]_{t2} = [OH]_{t1} \times e^{-(kOH+k_w) \times \Delta t} \qquad (3)$$

$$kOH = \left\{ \ln\left(\frac{[OH]_{t1}}{[OH]_{t2}}\right) \right\} \times sr_{CIMS} - k_w \qquad (4)$$

with $sr_{CIMS}$ being the scaling rate $sr_{CIMS} = \frac{1}{(t2-t1)}$ in s$^{-1}$. The total loss rate as measured by CIMS includes the OH loss by chemical reactions (kOH) as well as OH loss onto flow tube walls ($k_w$). The wall loss rate can be quantified using Eq. (4)

when no OH reactants are delivered to CIMS via zero air, i.e. a zero measurement. The scaling rate $sr_{CIMS}$ is determined under synthetic conditions when a known amount of OH reactivity is delivered to the instrument (Section 2.3.)

To achieve the measurements of two OH concentrations on the OH decay curve, two consecutive measurements in two separate titration zones in the CIMS flow tube are implemented (Fig. 1). The distance between the titrations zones have been optimised for the physical and operational set up at the MOHp and a typical low kOH regime. The CIMS has a vertically

oriented inlet with a sample tube that is 0.63 m long to draw sample air through the observation platform floor into the laboratory where the instrument is installed at the ceiling. The injectors for the second titration zone were permanently installed 0.15 m downflow of the first zone which is at the tip of the sample tube. Considering the fixed sample mass flow rate, the distance between the two titration zones is equivalent to about 0.1 s reaction time. Typical CIMS OH reactivity measurement properties and system parameters are summarised in Table 1.

**2.3 OH reactivity calibration to determine scaling rate $sr_{CIMS}$ and wall loss rate $k_w$**

The scaling rate $sr_{CIMS}$ and wall loss rate $k_w$ are parameters that need to be determined experimentally by regular OH reactivity calibration. OH reactivity ranging from $0 - 20$ s$^{-1}$ is produced using propane as OH reactant from either home-made mixture of 0.18 % propane in $N_2$ (purity 99.999 %), or commercially available 0.2 % propane (purity 99.999 %) in $N_2$ (purity 99.9999 %) (Riessner-Gase, Lichtenfels, Germany). The OH reactant is delivered in a humidified synthetic air matrix

to the CIMS instrument by positioning an external glass flow tube on top of the sample inlet, schematically shown in Fig. 2. The humidified synthetic air matrix is made of Milli-Q ultrapure water, with a conductivity of 0.055 μS cm$^{-1}$ at 25 °C and up to 2 ppb Total Organic Carbon, and synthetic air (20 % $O_2$, 80 % $N_2$) with a purity of 99.5 % for $O_2$ and 99.999 % for $N_2$. The lower section of the glass tube that is illuminated by the UV light is made of Suprasil quartz to allow transmission of the 185 nm UV wavelengths and hence OH production in front of the inlet tip.

To obtain the scaling rate, CIMS OH measurements in titration zone 1 (T1) and titration zone 2 (T2), i.e. $\ln([OH]_{T1}/[OH]_{T2})$, are regressed against the known OH reactivity (Fig. 3). The regression slope itself constitutes the reaction time $\Delta t$ in seconds.



The scaling rate $sr_{CIMS}$ is the inverse of the reaction time in seconds. The wall loss rate $k_w$ is the product of the scaling rate and the value of $\ln([OH]_{T1}/[OH]_{T2})$ measured during the addition of zero OH reactivity, i.e. synthetic air only.

## 2.4    Facsimile model representation of OH reactivity measurement by CIMS

To investigate and simulate OH reactivity measurements by the CIMS instrument in different operating and chemical
conditions, radical and trace gas concentrations in the CIMS sample flow tube have been modelled. For this, a zero dimensional (0D) model has been used where the time evolution of ~1 second represents the development of trace gas concentrations within the flow tube from point of the inlet to the entry point of the vacuum system of the mass spectrometer downstream. The numerical simulations were executed using the FACSIMILE program (Chance et al., 1977), including several sulfur oxidation steps occurring during the conversion of OH to $H_2SO_4$ in the chemical mechanism scheme (see
Supplementary Material S1).

A typical modelling simulation involves initialisation with a given OH and $HO_2$ concentration representing their production in the UV zone in front of the flow tube. When modelling runs are to represent typical CIMS operating conditions, the absolute OH and $HO_2$ concentration produced in the UV zone, i.e. $OH_{\_UV}$ and $HO_{2\_UV}$, is not critical, however the ratio between $OH_{\_UV}$ / $HO_{2\_UV}$ can significantly affect the subsequent chemical cycling. For simulations representing typical
operational conditions, a ratio of $OH_{\_UV}$ / $HO_{2\_UV}$ = 1 is used, based on (R1) and (R2). All other relevant trace gases (e.g. propane for OH reactivity and NO to investigate HOx recycling effects) are also initialised. Then a sequence of 4 runs is done where the titration (= signal) and background modes (see Section 2.1 and Fig. 4) in the two titration zones are modelled to reproduce the individual OH signals that make up a modelled CIMS OH reactivity value. The scaling rate, i.e. the times of $SO_2$/propane injection in the model, and OH wall loss rate are prescribed based on laboratory based experimental values. The
injections were achieved in the modelling run by simply setting the concentration to the given level at the specific time step, ignoring any mixing time that might actually occur in the flow tube.

For analysis the time series of the radical and trace gas species were inspected over 1 s, $[OH]_{T1}$ and $[OH]_{T2}$ calculated from endpoint $H_2SO_4$ concentrations and modelled CIMS OH reactivity values determined. Where simulations were run to investigate $H_2SO_4$ production pathways with and without significant HOx recycling, all pathways that produced $H_2SO_4$
through recycled OH were tagged, allowing the separation of instrument internally produced OH and OH produced in the UV zone, i.e. initial or external OH.

## 3    Sources of uncertainty in CIMS OH reactivity measurements

In order to obtain OH reactivity values from the measured OH concentrations, Eq. (4) is evaluated and therefore uncertainties in the scaling rate $sr_{CIMS}$ and wall loss rate $k_w$ contribute to the uncertainty in OH reactivity. Here, the sources
of uncertainties for each parameter are considered.



### 3.1 Ambient pressure and temperature dependent reaction time

A fixed mass flow in the sample tube is maintained by a constant mass flow difference induced in the ionisation region (Berresheim et al., 2000) which means that with changing ambient temperature and pressure, the volume flow through the sample tube varies. This variability in volume flow affects the residence time in the tube, and therefore the OH reaction time

with OH reactants through changing the residence time between CIMS titration zone 1 and 2. Additionally, changes in volume flow rates can also affect flow characteristics in the sample tube, and therefore OH wall loss and mixing of injected $SO_2$. This constitutes a source of error which needs to be taken into account, especially so for long-term measurements where temperature and pressure have periodic behavior over a range of timescales.

Figure 5 shows the changes in reaction time calculated for F1 to F2 (Fig. 1) purely based on the observed temperature and

pressure changes, and therefore volume flow rates in the sample tube, at the Meteorological Observatory Hohenpeissenberg for the year 2015. A clear annual cycle is evident with smaller scaling rate $sr_{CIMS}$ (= longer reaction times) in the winter months and larger scaling rates $sr_{CIMS}$ (= shorter reaction times) in the summer. Additionally variability at shorter timescales is superimposed at daily and several day long cycles as a result of boundary layer dynamics, surface energy exchange and frontal weather systems. If the scaling rate was assumed to be a fixed value, as in the example of Fig. 5 a flow-based mean

$sr_{CIMS}$ of 6.0 s$^{-1}$ for 2015, the associated uncertainty of $\pm$ 0.17 s$^{-1}$ (1 sigma) and a maximum deviation at the order of $\pm$ 0.4 s$^{-1}$ would result in a non-negligible, seasonally systematic error when dealing with long-term data.

In order to account for this variability in CIMS OH reactivity reaction time, the long-term measurements at MOHp are calculated using a normalised scaling rate and concurrent temperature and pressure observations (Eq. 5 and 6). The scaling rate determined by laboratory experiment is referenced to the long-term climatological mean pressure and temperature at

MOHp ($T_{reference}$ = 7 °C, $p_{reference}$ = 904 hPa).

$$N_{TP} = \left(\frac{T_{reference}}{T_{observed}}\right) \times \left(\frac{p_{observed}}{p_{reference}}\right) \qquad (5)$$

$$kOH_{corr} = N_{TP} \times \left\{\ln\left(\frac{[OH]_{t1}}{[OH]_{t2}}\right)\right\} \times sr_{CIMS} - k_w \qquad (6)$$

### 3.2 Experimental determination of scaling rate $sr_{CIMS}$ and wall loss rate $k_w$

The uncertainties of the scaling rate $sr_{CIMS}$ and wall loss rate $k_w$ are dependent on the accuracy and precision of the calibration points (Fig. 3). The calibration of CIMS OH reactivity is on the one hand fundamentally based on knowing the absolute OH reactant concentration and its OH reactivity based on published kinetic rate constants. On the other hand, the uncertainty in the CIMS measured term $\ln([OH]_{T1}/[OH]_{T2})$ is e.g. dependent on CIMS instrument, UV-source and mass flow controller performances and quality of gas mixtures such as $SO_2$ mixture for OH titration.



### 3.2.1 Uncertainty in $\ln([OH]_{T1}/[OH]_{T2})$

There are several factors that can produce an error in the OH (i.e. $H_2SO_4$) concentration detection by CIMS during the OH reactivity measurements. These include conditions that change between the measurements in titration zone 1 and 2, such as e.g. $[H_2O]$, UV lamp intensity, wind speed and direction (i.e. crosswinds in front of the sample tube), all affecting $[OH]_{UV}$

production, as well as short term instrumental variability. Variability in ambient OH is expected to have a negligible effect because $[OH]_{UV}$ is larger by two orders of magnitude compared to typical variability in ambient OH. All these uncertainties are not dealt with here independently but are estimated by the standard deviation of consecutive or repeated measurement points.

During the OH reactivity calibration experiments the measurement repeatability in $\ln([OH]_{T1}/[OH]_{T2})$ is considered in the

linear regression determining the scaling rate (Fig. 3), and thus it also constitutes a factor in the uncertainty of the scaling rate itself. The repeatability in $\ln([OH]_{T1}/[OH]_{T2})$ is defined as the standard deviation of all replicate measurements at a given OH reactant concentration. Under synthetic conditions (such e.g. OH reactivity calibration), the median variability for $[OH]_{T1}$ was found to be 1.5 % and variability for $[OH]_{T2}$ was 80 % larger with a median of 2.7 % . However in absolute terms, the variability is larger for $[OH]_{T1}$ as it is on average larger than $[OH]_{T2}$ by a factor of four.

During the routine operation of ambient sampling at MOHp, five consecutive measurements are made in each titration zone, each taking 30 seconds, and each titration zone is sampled with and without the UV lamp on, i.e. sampling of each titration zone is five minutes long. The variability in the consecutive measurements as expressed by the standard deviation provides the basis for the uncertainty in the averaged values ($[OH]_{T1}$, $[OH]_{T2}$) that is used for the calculation of OH reactivity. Titration zone measurements in zone 2 immediately follow those in zone 1 so that the measured OH reactivity is effectively

an average over a ten minute period with an OH reactivity value provided every 20 minutes during routine operation. As an example for typical values at MOHp, looking at six months' worth of ambient OH reactivity measurements, both $[OH]_{T1}$ (mean = $118.9 \pm 5.8 \times 10^6$ molecules cm$^{-3}$) and $[OH]_{T2}$ (mean = $33.2 \pm 1.7 \times 10^6$ molecules cm$^{-3}$) have a mean error of 5 % (4.9 and 5.1 % respectively). The propagated error in the mean in the term $\ln([OH]_{T1}/[OH]_{T2})$ is 7.1 %.

### 3.2.2 OH reactant concentration in OH reactivity calibration gas mixture

Depending on the experimental set up for the OH reactivity calibration, OH reactant concentrations are either measured or calculated. In the case of parameters established during the chamber based OH reactivity comparison campaign (Fuchs et al., 2017) OH reactant concentrations were directly measured. In case of the standard OH reactivity calibrations at MOHp the OH reactant concentrations are calculated based on mass flow, concentration of the OH reactant gas mixture and dilution in the carrier gas. The accuracy of the OH reactant concentration in the gas mixture is considered here for a home-made

mixture of 0.18 % propane, a commercial 0.218 % propane mixture and a commercial 1 % CO mixture.

Both propane OH reactivity calibration mixtures were analysed for their propane content by the MOHp GC-FID system used for long-term measurements of VOC (Plass-Dülmer et al., 2002). To allow analysis with this system, the samples were




diluted by a factor of $10^6$. Propane concentrations were observed 12 % ± 6 % higher than declared on the label for the home-made and commercial mixture respectively, but agree within two standard deviations of the combined measurement error. A 12 % increase in mean concentration corresponds to an increase of 12 % of the scaling rate, and the 1 standard deviation measurement uncertainty (i.e. concentration of ± 0.011 % propane in mixture) is equivalent to about ± 5 ms uncertainty in

the reaction time. The measured propane concentrations were used to calculate OH reactivity with the concentration measurement uncertainty translating to an OH reactivity uncertainty of 5 %.

The commercial CO 1 % mixture (purity 99.5 % CO in 99.9999% $N_2$) has a specified 2 % measurement uncertainty, which means the relative error in calculated OH reactivity from this uncertainty is 2 %.

### 3.2.3 OH reactant contamination in calibration gas mixture

Whilst there is an uncertainty in the concentration of the chosen OH reactant, there is a potential additional error as a result from other OH reactants traces, i.e. contaminant OH reactants, in the calibration gas mixture. If the OH reactant contamination, other than NO, is large and unaccounted for, the scaling rate can be underestimated because the calculated OH reactivity is underestimated. In case of NO contamination HOx recycling inside the CIMS instrument can produce an underestimation of the $\ln([OH]_{T1}/[OH]_{T2})$ term and thus produce an overestimation of the scaling rate. For the case of the

0.18 % and 0.218 % propane mixture, no OH reactant contamination was found when a dilution of the mixture was analysed for 40 VOC species (Plass-Dülmer et al., 2002; Hoerger et al., 2015). This sampling approach required a dilution by a factor of $10^6$ which precludes a sensible estimation of an upper limit of OH reactivity from contaminant species. The measurements do not reveal substantial contamination, yet extra OH reactivity from contaminant traces cannot be ruled out.

### 3.2.4 OH reactant contamination in synthetic air (carrier gas)

CIMS calibrations of OH reactivity are done using synthetic or zero air as carrier gas. Uncertainty due to OH reactants traces in synthetic air do not affect the derivation of the scaling rate, i.e. the slope of the regression (Fig. 3) because a constant flow of synthetic air is used in experiments and any additional OH reactivity from carrier gas contamination provides a fixed OH reactivity across the whole range of calibration OH reactivity. However the wall loss rate can be affected as the measurement of zero reactivity is done with synthetic air, in which case contaminant OH reactants can lead to an overestimation of the

wall loss rate. To obtain an estimate of typical OH reactant contamination and consequently error in the wall loss rate synthetic air has been measured for a range of VOC and OH reactivity calculated. This standard synthetic air (Riessner-Gase, Lichtenfels, Germany) is composed of 20 % oxygen (purity 99.5 % $O_2$) and 80 % nitrogen (purity 99.999 % $N_2$).

The synthetic air was analysed for 40 VOC species (Plass-Dülmer et al., 2002; Hoerger et al., 2015) leading to a calculated OH reactivity of 0.02 ± 0.02 $s^{-1}$. It was observed that extensive flushing of tubing, valves etc. was required to remove trace

contamination before the zero OH reactivity measurements.

For the long-term OH reactivity measurements, not all synthetic air compressed gas cylinders used for zero measurements and OH calibrations were analysed for its contamination by VOC and inorganic OH reactants. Based on CIMS




measurements, the variability in the zero for the same gas cylinder, i.e. reproducibility of zero, is better than 6 %. However it has also been observed that differences between synthetic air cylinders can vary greatly in its contaminant levels. Differences in the CIMS zero measurement for different synthetic air cylinders have been observed to vary up to 42 %, indicating OH reactivity by contaminant OH reactants can make up to about 5 $s^{-1}$, making such cylinders unsuitable for OH reactivity

calibration or other experiments.

To address the potential source of error coming from using different synthetic air gas cylinders for the regular consecutive OH reactivity calibrations and zero measurements required for the long-term measurements, a reference synthetic air cylinder has been designated recently which is used for cross-referencing and quality control of synthetic air used in experiments.

**3.2.5 NO contamination in all gas mixtures**

It is important to consider the contamination of nitric oxide (NO) in all gas mixtures used for CIMS measurements due to its critical role in HOx recycling within the CIMS sample flow tube. NO can introduce significant measurement artefacts and therefore its presence in the gas mixtures used for OH reactivity calibration and routine measurements needs to be considered.

For this the Facsimile model has been used to investigate the effect of NO contamination on the determination of the scaling rate during OH reactivity calibration. Then a modelled scenario of no NO presence in the system is compared with a scenario of a realistic contamination in the experimental set up. The following realistic nominal values have been used in the NO contamination case: 140 ppt NO from the $SO_2$ titration gas mixture and 20 ppt NO from the synthetic air carrier gas. Contamination of NO in the propane calibration gas mixture is considered to be negligible as a result of the high purity of the

gas mixture and the applied dilution, leading to sub ppt NO concentrations in the sample flow.

The effect of NO on instrument internal HOx recycling is non-linear and dependent on both the level of NO present and the magnitude of OH reactivity, which is discussed in detail in Section 3.4. In the case of the OH reactivity calibration, when considering OH reactivity of 0-40 $s^{-1}$, the modelled NO contamination case overestimates the scaling rate by 5 %. For OH reactivity 20-40 $s^{-1}$, overestimation increases to 9 %. When considering the range from 0-20 $s^{-1}$ for the OH calibration

regression (Fig. 3), which is typically seen in ambient air at MOHp, the effect of NO contamination on the scaling rate is less significant: the overestimation of $sr_{CIMS}$ is 1 %. This shows that the determination of $sr_{CIMS}$ in cases of NO contamination of the order of tens to hundreds of ppt NO is more accurate when small OH reactivity is considered (0-20 $s^{-1}$). Thus, paying particular attention to NO contamination and minimising its effect on the scaling rate determination forms part of best practice for this OH reactivity measurement method. This is also relevant in terms of long-term stability of OH reactivity

measurements as the NO contamination in the 2 % $SO_2$ gas mixtures can vary. Concentrations of NO in the CIMS sample flow from different $SO_2$ cylinders have been found to range from the order of tens of ppt to up to 380 ppt NO. The 2 % $SO_2$ gas cylinder normally needs to be replaced every 3-6 months and the current standard operating procedure includes the measurement of NO in each $SO_2$ mixture to track potential sources of error from NO contamination.



### 3.2.6. OH kinetic rate constants

Recommended rate constants from both IUPAC (Atkinson et al., 2006) and JPL assessments (Burkholder et al., 2015) are provided with uncertainty factors, which are considered here. Discrepancies coming from the difference between recommendations are assessed for cases CO and $C_3H_8$ as OH reactants and described in detail in Supplementary Material S2.

The uncertainty in the scaling rate from the IUPAC kinetic rate constant uncertainty (1 sigma) is for 15 % for CO and 2 % for propane, which highlights that difference in uncertainties of rate constants can be one point of consideration when choosing an OH reactant for CIMS OH reactivity measurement calibration. This issue will also be relevant for any other method that requires absolute OH reactivity values for calibration. Here, the uncertainty in the rate constant is included in the uncertainty estimation of calibration parameters such as the scaling rate $sr_{CIMS}$.

### 3.3 Validation of CIMS OH reactivity calibration using external glass flow tube

The CIMS OH reactivity calibration method using an external glass flow tube was developed and designed around the need of ease of use, minimising disturbance and downtime for the continuous operational CIMS measurements (OH, $H_2SO_4$) as well as remaining as close as possible to the normal measurement conditions with respect to e.g. turbulence at tip of the inlet and flow characteristics in the sample flow tube. The validity of the OH reactivity calibration method as used at the MOHp

was confirmed when the CIMS instrument took part in the first comprehensive OH reactivity instrument comparison campaign at the atmospheric SAPHIR chamber at the Forschungszentrum Jülich (FZJ), Germany, in April 2016 (Fuchs et al., 2017). At the beginning of the comparison campaign an OH reactivity calibration using the external flow tube and propane as OH reactant was carried out (Table 2). During the campaign the CIMS operating conditions were the same as at MOHp, i.e. the total air inlet flow (Fig. 1) and hence sampling rate from the chamber was 2280 L/min. This high flow rate

meant that the chamber was operated in a flush mode and the concentrations of any injected OH reactants diminished rapidly within a few hours as a result of continuous dilution with synthetic air (Fuchs et al., 2017). It was possible to utilise four chamber experiments for the validation because the chamber roof was closed (excluding photochemistry in the chamber) and no NO or other reactive trace gases such as ozone was present in the chamber. CO was used as OH reactant in two experiments and pentane and a mixture of monoterpenes were each used once. For CO the determination of OH reactivity

was done using the CO concentration measurements in the chamber. For pentane and monoterpenes, OH reactivity measurements from the LIF instrument from the Forschungzentrum Jülich (FZJ) are used for the determination of the CIMS scaling rate due to the low time resolution of the VOC measurements. The FZJ LIF OH reactivity measurements have been shown to be highly accurate and precise (Fuchs et al., 2017) thus providing reliable absolute OH reactivity values for this validation.

The temperature and pressure normalised scaling rates for the 6[th] (external flow tube) and 7[th] April 2016 (chamber, Table 2) agree with each other within one standard deviation. This agreement thus confirms that the OH reactivity calibration approach with the external flow tube (Fig. 2) is a valid and applicable method. Comparing all $sr_{CIMS}$ values obtained during



the chamber campaign in April 2016, it is evident that there is some variability which cannot be explained by the quoted uncertainty and the maximum difference in the mean is 0.9 s$^{-1}$. Similar differences have been observed for calibrations carried out at MOHp over a period of 6 months, all using the same calibration gas mixture, showing a maximum difference in the mean sr$_{CIMS}$ values of 1.1 s$^{-1}$. These differences observed during both the chamber campaign as well as MOHp
calibrations point toward a significant source of uncertainty which has not been understood yet. In terms of the mean values for Jülich chamber campaign and MOHp, the normalised sr$_{CIMS}$ agree within one standard deviation uncertainty, indicating the long-term stability and robustness of CIMS measurement system.

### 3.4. Upper OH reactivity measurement limit

The upper measurement limit for the described CIMS system is largely determined by the detectable OH concentration in
titration zone 2. As the second set of injectors is at a fixed position in the CIMS sample tube, the reaction time is therefore only variable by the flow rate. Increasing the sample flow rate would decrease the transit and reaction time between titration zone 1 and 2, leading to an increase in the upper measurement limit as detectable OH concentration could be measured at higher OH reactivity. However for a fixed (mass) flow rate as used in this system, the upper measurement limit is therefore dependent on the initially produced [OH]$_{UV}$ concentration (R1). With the current instrument setup (Table 1), the upper
detection limit is about 40 s$^{-1}$. At this high end of OH reactivity HOx recycling in the presence of NO can introduce an additional uncertainty and determining the effective upper measurement limit under those conditions is discussed below.

To investigate CIMS OH reactivity influencing factors, the Facsimile model has been run for CO as OH reactant, contrasting effects on CIMS OH reactivity with and without an impurity of NO. An impurity of NO can be present in the commercial SO$_2$ gas mixtures and its effect on measurements and uncertainties is herewith assessed. For the model runs a mixing ratio of
140 ppt NO in the sample flow is used and was added into the model run at the same time as the SO$_2$ injection for OH titration. OH reactivity up to about 60 s$^{-1}$ has been modelled and model set up parameters (temperature, pressure, water vapour concentration, initial [OH]$_{UV}$ and [HO$_2$]$_{UV}$, CIMS scaling rate and wall loss rate) have been based on typical observed conditions. The reasons why CO was chosen here is that it is a simple compound that is used for OH reactivity calibrations in general (Wood and Cohen, 2006), the reaction of CO+OH leads to the direct production of HO$_2$, and NO induced HOx
recycling in the absence of any RO$_2$ presents the simplest case of HOx recycling.

The model was run with the OH recycling tagged version of the chemical mechanism (see Supplementary Material S1) whereby the total H$_2$SO$_4$ signal can be separated into the contributions coming from OH$_{UV}$ (H$_2$SO$_4$_OH), from recycled OH (H$_2$SO$_4$_OHrec) and from reaction chain HO$_2$+SO$_2$ (H$_2$SO$_4$_HO$_2$) and from reaction chain RO$_2$ + SO$_2$ (H$_2$SO$_4$_RO$_2$). Absolute H$_2$SO$_4$ concentrations for each pathway, relative contributions to total H$_2$SO$_4$ signals and differential or final
H$_2$SO$_4$ signals (i.e. titration mode measurement (signal) minus background mode measurement (bkg), Fig. 4) have been analysed (Fig. 6 and 7).

Figure 6 shows the modelled H$_2$SO$_4$ concentrations for both titration zones and for the runs with and without NO impurity. It is evident that recycled OH contributes to the final H$_2$SO$_4$ concentration even in the case without NO (black dashed lines Fig.



6); however it is negligible because at the order of $10^3$ molecules cm$^{-3}$ it is several orders of magnitude smaller than the final $H_2SO_4$. This HOx recycling happens via SOx chemistry only, and is not very efficient as the reaction $SO_2 + HO_2$ is slow with an upper limit rate of 1.0 x $10^{-18}$ molecules cm$^{-3}$ s$^{-1}$ (Atkinson et al., 2004).

Whilst all the recycled OH in the no impurity NO case comes from the SOx chemistry itself, HOx recycling in the presence
of NO can proceed faster. Although $[SO_2] > [NO]$ by about five orders of magnitude, the OH recycling reaction $HO_2 + NO$ is about seven orders of magnitude faster than the reaction $HO_2 + SO_2$. Therefore trace levels of NO do have to be taken into account for CIMS OH reactivity measurements.

In the case of the NO impurity, the $H_2SO_4$ from the recycled OH is increased by two orders of magnitudes (blue dashed lines Fig. 6). For titration zone 1, this remains a negligible proportion of the final $H_2SO_4$ concentration, but for titration zone 2 this
is no longer the case. For OH reactivity larger than 30 s$^{-1}$, remaining $OH_{UV}$ concentration that is to be measured is of the same order as the recycled OH. The contribution from the recycled OH leads to a deviation from the exponential first order decay (Fig. 6b, blue solid line), and thus to an overestimation of the $H_2SO_4$ concentration, implying less OH loss than expected for a given OH reactivity, and consequently CIMS OH reactivity above ca. 30 s$^{-1}$ is underestimated. At an OH reactivity of 40 s$^{-1}$ this underestimation is 5 s$^{-1}$ in the modelled NO impurity case.

The CIMS measurement cycle with signal and background modes is designed to cancel out contributions to artefact $H_2SO_4$ signals. However in the case of HOx recycling within the titration zone, this is not the case because recycled OH in the background mode is smaller than that in the signal mode (Fig. 7a). This is true for both titration zones, but of relevant magnitude in zone 2 only. This means that the signal from the recycled OH in the titration is not annulled by the background measurement, remaining an artefact contribution to the final $H_2SO_4$ concentration that is used for calculating OH reactivity
(Fig. 7b).

The relevance of the recycled OH in titration zone 2 becomes obvious inspecting Fig. 7c which reveals that in the case of NO presence, at increasing OH reactivity (> 40 s$^{-1}$) when remaining $OH_{UV}$ becomes smaller (solid blue line), the relative importance of recycled OH increases (dashed blue line), ultimately dominating the total $H_2SO_4$ concentration, rendering the CIMS OH reactivity measurement invalid for those conditions. The non-linear effect on final $H_2SO_4$ from HOx recycling
compared to the NO-free case is also illustrated in Fig. 7d, showing that when NO is present, the concentrations increase non-linearly with increasing OH reactivity, and significantly more so in the background mode than in the signal mode.

In conclusion, HOx recycling in the presence of trace level NO (here as example 140 ppt is used in the model) is relevant at high OH reactivity and therefore needs to be considered when choosing the range of OH reactivity for calibration (Section 2.3). To minimise this potential systematic error during the OH reactivity calibration, scaling rates have been determined for
OH reactivity ranging from 0-20 s$^{-1}$. Using CO for OH reactivity of 20 s$^{-1}$ with 140 ppt NO, the modelled underestimation is 0.5 s$^{-1}$, i.e. 2.5 %.

An additional consideration for the upper measurement limit (without NO) is that the to-be-measured OH concentration (Fig. 7b, solid black line) becomes a small residual between two larger numbers in signal and background mode at increasingly



high OH reactivity. The measurement precision (51 %, Berresheim et al., 2000) thus can also impose a constraint on the upper measurement limit.

**3.5 HOx recycling in the presence of ambient air NO**

An integral part of OH detection by CIMS is the chemical conversion of OH to $H_2SO_4$ inside the flow tube. Under certain conditions within the flow tube, HOx recycling ($HO_2 + NO \rightarrow NO_2 + OH$ (R6)) can become significant enough between the two titration zones and produce artefact signals of $H_2SO_4$ which are evident for measurements especially in the second titration zone. The $HO_2$ in the flow tube can come from ambient air and also be instrument internally produced through production in the UV zone (R1, R2) and be a product of the titration sequence (R3, R4).

HOx recycling in the presence of nitric oxide (NO) originating from ambient air or impurities in CIMS operational gas mixtures can lead to an apparent underestimation of OH reactivity (Section 3.4). At the MOHp, the site for which the CIMS instrument has been optimised, NO levels rarely exceed 4 ppb (99[th] percentile is 3.7 ppb) and the 95[th] percentile is 1 ppb NO. However in principle ambient NO concentrations can reach up to tens of ppb close to sources such as e.g. tailpipe emissions. Therefore, the interference of NO on CIMS OH reactivity measurements has been characterised through a series of laboratory experiments for mixing ratios up to 20 ppb NO. The experimental set up is schematically shown in Fig. 2 and replicates the OH reactivity calibration experiments with the addition of set concentrations of NO. The interference experiments were carried out for different OH reactants, representing different groups of reactants, i.e. carbon monoxide CO (inorganic specie), propane $C_3H_8$ (alkane), ethene $C_2H_4$ (alkene of mostly anthropogenic origin) and isoprene $C_5H_8$ (alkene of biogenic source). The effect of OH reactivity underestimation was considerably stronger for CO than for the other OH reactants, so only data from propane, ethane and isoprene experiments were used to derive a correction function that would be representative for an ambient air matrix. Measurements above 15 ppb NO were also excluded from the correction function analysis, as measurements had low reproducibility, the reasons for which are unclear.

As also expected from Facsimile model runs of OH recycling in the CIMS sample tube, the experimental results as seen in Fig. 8 show clearly that the underestimation of CIMS OH reactivity is non-linearly dependent on both NO concentration and OH reactivity. By exploring CIMS OH reactivity measurements for each NO concentration (Fig. 9a), it was found that the correction could be empirically well described by an exponential equation of the form

$$OH\ reactivity = a \times e^{(-b \times kOH_{measured})} + c \qquad (7)$$

where factors a, b and c are functions of NO concentration in ppb (Fig 9b-d). Factors a and b are exponential functions

$$a = aa \times e^{(-ba \times [NO])} + ca \qquad (8)$$

$$b = ab \times e^{(-bb \times [NO])} + cb \qquad (9)$$

Factor c is a linear function, described by



$$c = ac \times [NO] + bc \qquad\qquad (10)$$

To find the eight fit parameter aa, ba, ca, ab, bb, cb, ac and bc, a Python SciPy curve fitting procedure was applied to the experimental data (Jones et al., 2015). Marginally better fit results ($R^2$ = 0.98) were achieved when input values were averaged for the three different OH reactants (propane, ethene and isoprene) (Fig. 8) compared to individual experiments (e.g. $R^2$ = 0.95 for ethene only). Whilst the overall agreement of the fitted, i.e. corrected, CIMS OH reactivity shows little bias with OH reactivity with a slope of 0.98 (Fig. 10a), when looking at the residuals (Fig. 10b,c) it is apparent that there is a positive bias between 0-3 $s^{-1}$, and there is a negative bias between 3-10 $s^{-1}$. The positive bias below OH reactivity of 3 $s^{-1}$ has got a mean overestimation of the corrected OH reactivity of 1.2 $s^{-1}$, i.e. on average an overestimation of 95 %. The negative bias between 3-10 $s^{-1}$ is smaller and is on average 10 % underestimation. These biases highlight one of the disadvantages of this empirical fit function, pointing to a systematic process or effect that is not captured by this function. The fit parameters do not directly represent any physical or chemical processes or our understanding of the HOx recycling chemistry. Additionally the fit function represents an approximation of the actual recycling that is dependent on the concentration of $HO_2$ and $RO_2$ present in ambient air. One standard deviation in the residuals is +/- 1.5 $s^{-1}$ which represents a large relative error at very low OH reactivity. Therefore, at low OH reactivity (below ~3 $s^{-1}$) applying this correction function would introduce a significant additional error in the CIMS OH reactivity measurement. This was considered when the empirical NO correction function was applied to data from the OH reactivity instrument comparison campaign where OH reactivity reached the CIMS upper measurement limit and large concentrations of NO were at times present in the chamber. The correction was not applied for OH reactivity below 2.5 $s^{-1}$ (Fuchs et al., 2017), because the systematicity in error was found to be related to OH reactivity at the very low end of OH reactivity (Fig. 10c). No systematicity in errors related to NO concentration was identified.

One source of uncertainty lies in the reproducibility or robustness of the determined effect of NO on measured CIMS OH reactivity. To investigate how robust and representative the uncertainties in corrected OH reactivity values are, the distribution of residuals for different sets of experiments were checked. One dataset consisted of all individual points from the above mentioned isoprene, ethene and propane experiments carried out on the same day. This dataset includes the potential uncertainties from different $RO_2$ for the different OH reactants. The other dataset consisted of four different NO-HOx recycling experiments over the course of a year, using propane only and therefore giving an indication of the uncertainty associated with the reproducibility of the HOx recycling interference effect.

A normal distribution (Gaussian) fit function was applied to the data and fitted mean and standard deviation estimated (Fig. 11). Figure 11 shows that in both sets of experiments the bias in the mean correction is small (less than 0.42 $s^{-1}$), however the one standard deviation in the mean error of the fit is about ± 2.7 $s^{-1}$. This analysis shows that with the empirical eight parameter fit the uncertainty from using different OH reactants for the NO-HOx recycling experiments is of the same order (1 σ about ± 2.7 $s^{-1}$) as for the uncertainty from measurement and experimental stability, as tested by several experiments over the course of a year using one OH reactant, here propane, only.



In contrast to the range of OH reactivity and NO concentration during the characterisation experiments, the continuous long-term observations at the Meteorological Observatory Hohenpeissenberg show that the typical OH reactivity is below 15 s$^{-1}$ and NO well below 15 ppb. Instead of applying a NO-based correction function which introduces itself considerable uncertainty to the long-term data from 2009-2017, we take the approach to include a NO concentration based error

estimation in the total CIMS OH reactivity measurement uncertainty.

For OH reactivity up to 15 s$^{-1}$ and NO up to 4 ppb, the systematic underestimation in OH reactivity can be approximated by a linear relationship (Fig. 12). Figure 12 shows the mean underestimation in OH reactivity binned for all OH reactivity and NO concentration where the error bars represent the 1 sigma variability in measured OH reactivity underestimation for the given NO bin. The underestimation for MOHp-prevalent conditions is estimated as 0.8 s$^{-1}$ / ppb NO (Fig. 12, filled circles).

Extending the range of NO concentration to 15 ppb shows that this linear relationship approximately holds even for concentrations larger than 4 ppb NO. Using this error estimation approach for the long-term data at MOHp allows the determination of expected underestimation of measured OH reactivity based simply on ambient NO concentration measurements. This systematic error is then included in the total uncertainty estimation on a point by point basis. As the 95[th] percentile of NO mixing ratio at the MOHp is 1 ppb, the majority of data will have a positive error estimation of less than 0.8

s$^{-1}$.

## 4 Performance of CIMS in chamber and field studies

### 4.1 OH reactivity instrument comparison at FZ Jülich SAPHIR chamber

The CIMS instrument was part of a comprehensive OH reactivity instrument comparison campaign at the Forschungszentrum Jülich (FZJ) SAPHIR chamber in April 2016. The full findings and results from that campaign are

evaluated and discussed in Fuchs et al. (2017). With respect to CIMS, one main finding was that the CIMS instrument provides high precision data with a limit of detection better than 1 s$^{-1}$ at a time resolution of a few minutes. In chemically complex conditions, the CIMS data show a scatter of 10-20 %. The presence of up to 32 ppb NO in the SAPHIR chamber revealed limitations of the current CIMS system, and a NO-based correction function was applied to the CIMS data, applicable for OH reactivity up to 40 s$^{-1}$ and 15 ppb NO (details see section 3.5, Eq. 7). No data above those limits were

submitted to the comparison exercise described in Fuchs et al. (2017). The quality of the correction was shown to be variable, with OH reactivity being overestimated by a factor up to 1.8 for NO concentrations from 10-15 ppb. The NO eight parameter fit correction function was derived from laboratory experiments at MOHp, and fit parameters applied to the chamber dataset. The poor performance at higher NO concentrations shows that there is an issue of applicability of the function derived from experiments with a certain set of conditions (MOHp) to a dataset where conditions were different

(SAPHIR chamber). For instance, a simple modelled comparison shows that the 10 % higher atmospheric pressure leads to more effective NO-HOx recycling and therefore 10 % greater underestimation of the measured CIMS OH reactivity (prior to correction) would be expected at FZJ / 1000 hPa. This difference and potential error have not been considered in the





correction of CIMS OH reactivity in Fuchs et al. (2017). In future measurement campaigns away from the MOHp, the benefit of performing NO interference experiments during the campaign shall be considered.

It is worth highlighting that the fit parameter do not represent physical or chemical variables, and therefore the function does not easily lends itself to improvements or systematic interrogation when and why the correction fails. Further development
on a correction function is ongoing that includes measured and estimated variables such as $HO_2$, $RO_2$ and $HO_2$ wall loss in the CIMS sample tube.

The comparison including OH reactivity up to 40 s$^{-1}$ has shown that CIMS measurements were of high accuracy for certain chemical conditions (experiments with CO, pentane, monoterpenes, sesquiterpenes), but had a lower accuracy for others (isoprene, MVK, MACR mixture and urban mixture of o-xylene, toluene, 1-pentene).
The instrument comparison has also revealed that the CIMS provides high quality OH reactivity measurements for conditions of low OH reactivity (< 15 s$^{-1}$) and low NO concentrations (< 4 ppb). The particular performance within those ranges, which are typically observed at MOHp, is specifically assessed here. The FZJ LIF instrument was shown to be highly performant throughout the campaign, giving consistently high accuracy OH reactivity measurements (Fuchs et al., 2017). Therefore the comparisons within the range of typical conditions seen at MOHp are discussed here using the FZJ LIF as
reference.

To investigate the CIMS measurement quality for conditions at the MOHp, a subset of the data from the comparison campaign was produced. All campaign data (i.e. all experiments) were selected, averaged over 2 minutes intervals, and filtered for NO concentrations below 4 ppb and FZJ LIF OH reactivity below 15 s$^{-1}$. For these conditions, Fig. 13a shows this subset of data evaluated in Fuchs et al. (2017). The linear regression slope of 0.94 shows that the CIMS is typically
underestimating OH reactivity by 6 % compared to the LIF OH reactivity measurements. The error bars for the CIMS measurements are large (approx. ± 1.5 s$^{-1}$) because the total error includes the 1 standard deviation error in the fit residuals from the eight parameter NO correction (Fig. 10b). The NO correction was applied to all data as a result of NO contamination in the $SO_2$ titration gas mixture (Fuchs et al., 2017).

To assess the impact of not correcting for the presence of NO in the MOHp OH reactivity dataset explicitly but rather
account for the systematic underestimation by a positive error, CIMS OH reactivity from the SAPHIR campaign was reevaluated and total uncertainty include the error from the presence of NO. This constitutes an independent test whether this data treatment approach for the long-term OH reactivity measurements at the MOHp is acceptable. The regression slope is 0.95, showing an underestimation compared to LIF of about 5 % (Fig. 13b). This shows that for the conditions seen at MOHp, it can be considered acceptable to not explicitly correct for presence of NO, but represent the underestimation
through increased measurement uncertainty. For the comparison campaign propane was used for the scaling rate determination and it is worth noting that the difference in rate constant recommendations and rate constant uncertainties for propane (Supplementary material S2) is at the order of a few percent, and therefore the discrepancy between CIMS and LIF of 5 - 6% falls within the two sigma uncertainty of the OH + propane rate constant recommendations.





The regression in Fig. 13b shows a negative intercept of -0.63 s$^{-1}$ in CIMS data when one fixed wall loss rate is applied for the whole campaign. If however the zero values measured at the beginning of each day in the chamber (Fuchs et al., 2017) are used to estimate a daily wall loss rate, the intercept is -0.12 s$^{-1}$, indicating that regular zero measurements can improve data quality. The campaign wall loss rate used in Fig. 13 was based on the zero measurement (with 3 % error) of the day

before the comparison campaign, using standard synthetic air. A mere overestimation of 7 % in this zero value could cause the intercept of -0.6 s$^{-1}$. During repeatability tests in the laboratory at MOHp, differences between zero measurements tend to be below 7 %, but 6 % differences have been observed (Section 3.2.4). This underlines the point that high quality zero OH reactivity measurements contribute to achieving high-accuracy measurements.

## 4.2 Ambient OH reactivity measurements at MOHp

The Meteorological Observatory Hohenpeissenberg (MOHp) carries out continuous measurements of reactive gases, aerosols, radiation and meteorology as part of the Global Atmosphere Watch (GAW) programme (Schultz et al., 2015). Adoption of the integrative, holistic and direct measure of OH reactivity at long-term measurements sites such as those belonging to GAW holds the potential for insight into changes of oxidative capacity of the troposphere (Williams and Brune, 2015). We demonstrate here that these continuous measurements are possible.

The MOHp is situated in Southeast Germany, 40 km north of the Alps, at the top of a hill (985 m a.s.l.). The Hohenpeissenberg hill rises 300-400 m above the surrounding rural area with about 70 % mostly coniferous forest and 30 % agricultural pastures.

The main wind direction at the MOHp is South to Southwest and the advected air masses are generally clean and pollutant levels are low for these southern wind sectors (Mannschreck et al., 2004; Bartenbach et al., 2007). Air coming from the

Northeast tends to have higher pollutant levels as emissions from the large city of Munich (city population of 1.5 million, ~80 km Northeast from MOHp) increase atmospheric pollutant loadings. Pollutants with longer lifetimes, such as CO, are less variable with wind direction in contrast to shorter lived species such as NOx which show larger variability with wind direction (Mannschreck et al., 2004).

OH reactivity measurements at the MOHp started in 2009 and have been continuous since then. To our knowledge this

constitutes the longest time series of OH reactivity measurements to date. The standard OH reactivity measurement interval is 20 minutes and it forms part of a sequential measurement block of H$_2$SO$_4$, OH, ROx and OH reactivity. Here we present measurements from one year, 2015, exemplifying the continuity and stability of the long-term CIMS OH reactivity measurements at MOHp (Fig. 14). Error bars in Fig. 14 include the contribution of underestimation as a result of the presence of ambient NO, the uncertainty in the scaling rate and wall loss rate.

OH reactivity at MOHp is mostly below 15 s$^{-1}$ (Fig. 14) which is characteristic for this kind of environment and similar magnitudes were observed in mid latitude forests and also at some suburban sites (Yang et al., 2016). No clear annual cycle is visually identifiable for 2015 and variability occurs at a range of time scales. Absence of a strong annual cycle was also modelled based on OH reactant observations at MOHp for the years 1999-2003, with typical monthly OH reactivity between





3-4 s$^{-1}$ (Rohrer and Berresheim, 2006). The variability in OH reactant concentrations is dependent on a range of factors, including e.g. periodic and non-periodic dynamics (Huntrieser et al., 2005; Mayer et al., 2008). As for the periodic daily cycle, reactive trace gases are significantly controlled by the rise and fall of the boundary layer top passing the measurement height. The boundary layer is generally below the station at night, meaning that the nighttime sampling then occurs in the

normally more uniformly mixed residual layer. For cold and windy nighttime conditions however, this decoupling does not always occur, thus affecting the variability of pollutant concentrations (Bartenbach et al., 2007). The effect of the boundary layer building at the beginning of the day and reaching the MOHp can be observed e.g. by the peak concentrations of VOC in the morning (Bartenbach et al., 2007).

In addition to the dynamically driven effects on trace gas concentrations, short lived reactive species are strongly controlled

by local emissions and chemical cycling, such as e.g. for biogenic VOC, which in sum have been observed to have clear diurnal cycle at MOHp (Handisides et al., 2003; Bartenbach et al., 2007). In contrast, anthropogenic VOC have a less pronounced daily cycle (Handisides et al., 2003; Bartenbach et al., 2007) and dominance of biogenic VOC in summer is contrasted by highest concentration of anthropogenic VOC, such as ethane and propane, in winter (Helmig et al., 2016). Additionally, inorganic trace gases such as CO and $NO_2$ which contribute greatly to the total OH reactivity at MOHp also

show highest levels in winter. Shorter lived $NO_2$ even displays a marked weekday variation with highest concentrations midweek and lowest levels on Sundays (Gilge et al., 2010) and sharp short term peaks also do occur (Handisides et al., 2003; Acker et al., 2006). Winter shows greater short term variability in OH reactivity (Fig. 14), which is likely a result of more local pollution, especially from domestic wood burning for heating. This combined with the fact that persistent inversion layers can form in winter, contributing to enhanced mixing ratios (Gilge et al., 2010) can go some way to explain this short

term variability.

As addressed in Section 3, the total measurement uncertainty in OH reactivity depends on a variety of factors that contribute to systematic and random errors. To illustrate the typical total measurement uncertainty for ambient measurements at MOHp, errors from each term in the OH reactivity calculation (Eq. 4) are examined and their contribution to the total uncertainty assessed. Calculations and propagation of uncertainties was performed using Python Uncertainties package (Lebigot, 2017)

which predicts uncertainty using linear error propagation theory. The three terms are the point by point values of $\ln([OH]_{T1}/[OH]_{T2})$, the scaling rate $sr_{CIMS}$ obtained regularly by OH reactivity calibration and the zero air value of $\ln([OH]_{T1}/[OH]_{T2})$, also determined regularly. The mean error in $\ln([OH]_{T1}/[OH]_{T2})$ is 5 % (Section 3.2.1) and the upper limit of reproducibility of the zero air value of $\ln([OH]_{T1}/[OH]_{T2})$ is 6 % (Section 3.2.4). As for the scaling rate $sr_{CIMS}$, the uncertainty in the individual values of the scaling rate for the five different experimental days in the Jülich chamber in 2016

is below 2.4 %, however the variability between days is larger. One standard deviation in the campaign mean value of $sr_{CIMS}$ amounts to 5.3 % (Table 2). Using a ambient air representative example with $\ln([OH]_{T1}/[OH]_{T2}) = 1.28 \pm 0.07$, zero air = 1.0 $\pm$ 0.06, $sr_{CIMS} = 8.62 \pm 0.46$ results in an OH reactivity of 2.4 $\pm$ 0.8 s$^{-1}$. The one standard deviation of $\pm$ 0.8 s$^{-1}$ is about equally determined by the error in the measured ambient air $\ln([OH]_{T1}/[OH]_{T2})$ and the reaction rate constant $sr_{CIMS}$. The uncertainty in the measured zero air value of $\ln([OH]_{T1}/[OH]_{T2})$ is not dominant here. To decrease the total measurement




uncertainty addressing uncertainties in both $\ln([OH]_{T1}/[OH]_{T2})$ and $sr_{CIMS}$ will be effective, because the squares of the individual uncertainties is summed. In this example the positive systematic error from the NO interference (defined in Section 3.5 as 0.8 s$^{-1}$ / ppb NO) would only be increasing the total measurement error for NO concentrations above 330 ppt.

It is worth noting that the example given above is meant to represent operation in typical measurement conditions and the

total measurement uncertainty that can be expected with this method and instrument in its current set up. However it is clear that there are "known unknowns" that can affect the accuracy of the OH reactivity measurements. These are extensively discussed in Section 3, such as e.g. OH reactant contamination in synthetic/zero air and in the OH reactant gas used for calibration which impact the accuracy of the scaling rate and wall loss rate. In light of the continuous long-term operation of the instrument, eliminating, minimising and/or quantifying and accounting for these systematic errors forms part of the data

quality control protocol that has been successively developed for these measurements. The comparison of CIMS OH reactivity measurements with the FZJ LIF measurements (Section 4.1) shows that these errors are unlikely to be large however, proving that this semi-direct method to measure OH reactivity by CIMS provides robust and accurate observations.

## 5 Conclusions

We presented a novel approach to measure OH reactivity successfully by a semi-direct method using a chemical ionisation

mass spectrometer (CIMS) system. This approach is based on a well-established technique to measure OH and H$_2$SO$_4$ and provides an extension to the measurement portfolio at the Meteorological Observatory Hohenpeissenberg (MOHp) which also delivers continuous long-term observations of OH, H$_2$SO$_4$ and ROx by CIMS as part of the WMO Global Atmosphere Watch (GAW) programme. The development of the OH reactivity measurement came as an addition to the CIMS system measurement capability without losing existing measurements (OH, H$_2$SO$_4$) or compromising their quality.

OH is measured at the inlet tip by the conversion to H$_2$SO$_4$ using SO$_2$ as titrant, and an additional titration zone was added further down flow in the sample flow tube. This way one reaction time was achieved for OH reactivity measurements. The measurements require accurate determination of this reaction time as well as the OH wall losses in the flow tube.

Comparison of CIMS OH reactivity with measurements from a highly accurate and precise LIF system from the Forschungzentrum Jülich reveals that this new semi-direct CIMS method performs especially well for the conditions it was

designed for, i.e. low NO concentrations and low OH reactivity. For OH reactivity below 15 s$^{-1}$ and NO concentrations below 4 ppb, the mean underestimation of OH reactivity by CIMS was only 5 % for this comparison, which falls within the 2 sigma uncertainty of the rate constant recommendations required for the calculation of CIMS OH reactivity. Comprehensive assessment of the instrument performance for a wide range of chemical conditions can be found in Fuchs et al. (2017).

Continuous OH reactivity measurements at the MOHp started in autumn 2009 and a "snapshot" of one year of measurements

from 2015 were presented here. These unique measurements demonstrate that continuous long-term observations for OH reactivity are possible, enhancing the measurement portfolio at the global GAW site Hohenpeissenberg (MOHp).

©c Author(s) 2018. CC BY 4.0 License.





The method is described as semi-direct because OH is measured via the chemical conversion to $H_2SO_4$ which is then detected by CIMS and because only two points on the pseudo first order OH decay curve are measured. As a rule, the chemical conversion of OH to $H_2SO_4$ is an aspect of the system that needs to be considered for its potential sources of OH reactivity measurement error. A modelling study for instance showed that at high OH reactivity HOx recycling through the
titration SOx chemistry can lead to small systematic underestimation of OH reactivity. In the presence of NO, this instrument internal HOx recycling is enhanced, leading to significant underestimation also at lower OH reactivity. To minimise errors from this, it is therefore important to ensure the absence of or at least minimise NO and other contaminants in gases that are used to run the instrument, as well as during the OH reactivity calibration. It is recommended that concurrent ambient measurement of NO are made and that the titration gas ($SO_2$) be tested for NO contamination if its absence cannot be
guaranteed, as is often the case for commercially sourced mixtures for instance. To account for the unavoidable presence of NO in ambient air, it is critical to characterise the system-dependent response to the NO-HOx measurement artefact. For this CIMS system, a correction function was developed and tested in a recent OH reactivity instrument comparison campaign (Fuchs et al., 2017). The instrument comparison revealed limitations of the empirical function found to describe the systematic underestimation in OH reactivity which is both dependent on NO concentration and OH reactivity. For long-term
measurements at MOHp NO concentrations and OH reactivity are low (mostly < 1 ppb NO and < 15 $s^{-1}$) which do not warrant a correction for operational data, but a NO-based error is included in the data treatment to reflect the increased measurement uncertainty when NO is present in ambient air above a mixing ratio of few hundred ppt.

The NO induced HOx recycling reveals a challenge to the current method, which rests on the assumption that the OH decay in the sample tube is a pseudo first order decay. This assumption does not hold when OH recycling happens between the two
measurement points. To gain additional information on the OH decay curve, it would for instance be possible to add further titration zones. The current placement of the second titration zone was informed by the expected range of OH reactivity below 20 $s^{-1}$ at the MOHp. The fixed position of the second titration zone also imposes an upper limit to the OH reactivity that can be detected, which is around 40 $s^{-1}$ for this set up. The effective upper limit could be extended through increasing the flow rate through the sample tube and/or by installing titration injectors closer to the tip of the inlet.
Another approach to address the NO induced HOx recycling would be to change the production of OH and therefore effecting reduced OH recycling. For instance, it could be advantageous to produce only OH, and no $HO_2$, in front of the inlet, as e.g. through the use of $O_3$ flash photolysis by a 266 nm laser.

The method presented here shows that OH reactivity can be relatively easily measured with an existing CIMS instrument used for OH measurements. It provides accurate and continuous measurements in the specific conditions for which it was
developed, i.e. low OH reactivity and low NO concentrations typical for the MOHp site. If the system is to be used in other chemical conditions or environments, it is advised to make modifications to the set up or operation, as well as characterise and correct for systematic effects such as e.g. from NO induced HOx recycling.



## 6 Acknowledgements

We would like to thank Felix Utschneider, Katja Michl, Erasmus Tensing, Marita Hofmann and Eva Jettmar from DWD for technical support. We also acknowledge the generous support from staff of the "Photochemistry and Radicals" and "Reactive Trace Substances" departments of the Institute of Energy and Climate Research, IEK-8: Troposphere, at the Forschungszentrum Jülich, Germany. We are particularly grateful for their assistance during the OH reactivity instrument comparison campaign at the SAPHIR chamber and for the provision of the LIF OH reactivity campaign data.

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



**Table 1: MOHp CIMS kOH measurement and system properties.**

| CIMS system property | value and measure |
| --- | --- |
| [OH] limit of detection | $5 \times 10^5$ molecules cm$^{-3}$ |
| kOH limit of detection | $0.5$ s$^{-1}$ |
| kOH upper measurement limit | $40$ s$^{-1}$ |
| typical [OH]$_0$ and [HO$_2$]$_0$ produced in UV zone | $10^8$ molecules cm$^{-3}$ |
| kOH temporal resolution | $60 - 300$ s |
| typical reaction time $\Delta t = (t_2 - t_1)$ | $0.1$ s |
| typical OH wall loss rate $k_w$ | $10$ s$^{-1}$ |
| kOH measurement accuracy ($1\,\sigma$) | $1$ s$^{-1}$ (kOH $< 30$s$^{-1}$) |
|  | $2$ s$^{-1}$ (kOH 30-40 s$^{-1}$) |
| kOH measurement precision ($1\,\sigma$) | $0.14$ s$^{-1}$ |
| known interference | nitric oxide (NO) |





**Table 2: Summary of scaling rate sr$_{CIMS}$ and wall loss k$_w$ parameters determined from OH reactivity instrument comparison campaign. All sr$_{CIMS}$ values have been normalised using temperature and pressure to allow comparison between Jülich and MOHp datasets.**

| OH reactant | OH reactivity (kOH) used for calibration | normalised scaling rate sr$_{CIMS}$ ± 1 σ (s$^{-1}$) | measured zero ln([OH]$_{T1}$/[OH]$_{T2}$) ± 1 σ |
|---|---|---|---|
| Jülich SAPHIR chamber (April 2016) | | | |
| Propane 6.4.2016 | External flow tube | 9.6 ± 0.2 | 0.97 ± 0.02 |
| CO 7.4.2016 | Chamber concentration measurements | 9.4 ± 0.1 | 0.95 ± 0.03 |
| Pentane 8.4.2016 | FZJ LIF kOH measurements | 9.1 ± 0.1 | 0.89 ± 0.05 |
| Monoterpenes[a] 13.4.2016 | FZJ LIF kOH measurements | 8.8 ± 0.1 | 0.92 ± 0.11 |
| CO 15.4.2016 | Chamber concentration measurements | 8.7 ± 0.1 | 0.95 ± 0.06 |
| *Jülich mean ± 1 σ* [b] | | *9.1 ± 0.4* | *0.94 ± 0.03* |
| MOHP (July – December 2016) | | | |
| Propane 18.7.2016 | External flow tube | 10.1 ± 0.4 | 0.88 ± 0.14 |
| Propane 12.8.2016 | External flow tube | 9.5 ± 0.3 | 0.94 ± 0.04 |
| Propane 18.8.2016 | External flow tube | 10.1 ± 0.1 | 1.08 ± 0.02 |
| Propane 24.11.2016 | External flow tube | 9.5 ± 0.2 | 0.97 ± 0.05 |
| Propane 7.12.2016 | External flow tube | 9.0 ± 0.7 | 0.96 ± 0.05 |
| *MOHp mean ± 1 σ* [b] | | *9.7 ± 0.5* | *0.97 ± 0.07* |

[a] α-pinene, limonene, myrcene

[b] 1 σ is the standard deviation in the mean from all days



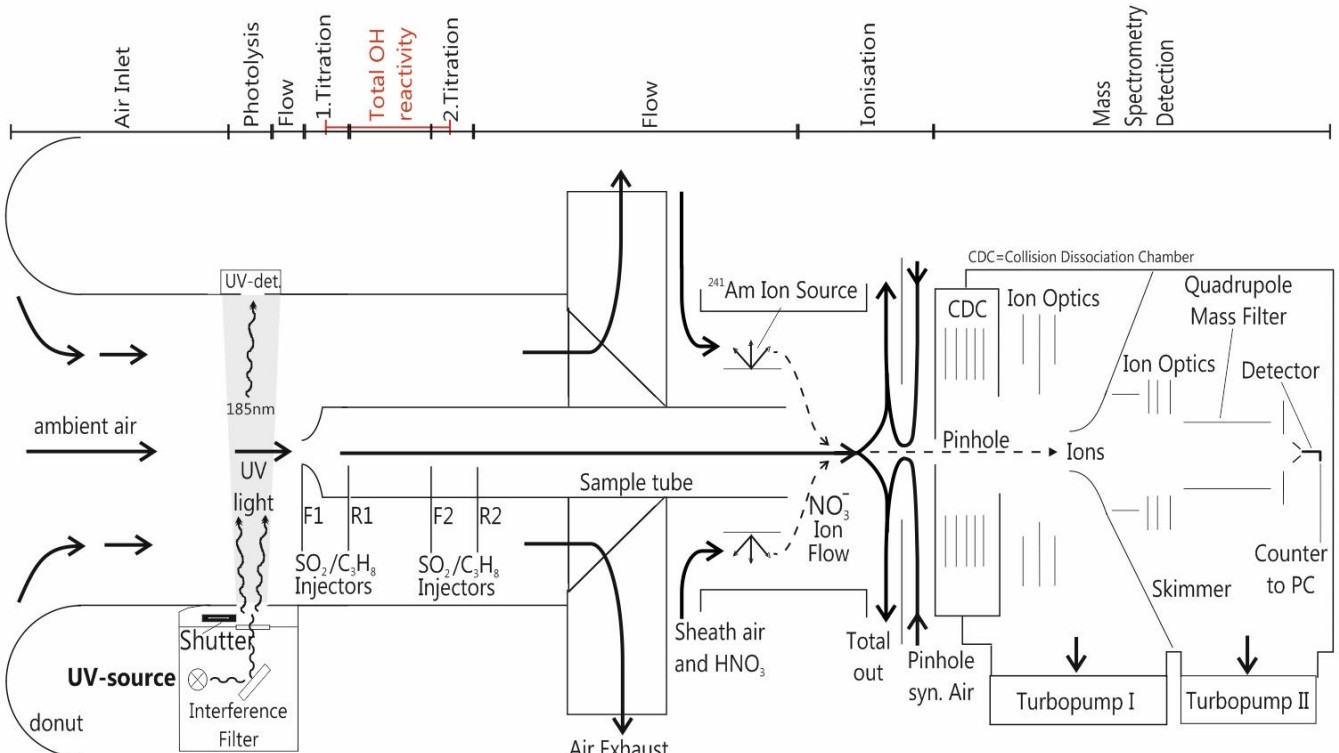

**Figure 1: Schematic of CIMS system used for long-term OH, H$_2$SO$_4$ and OH reactivity measurements at the Meteorological Observatory Hohenpeissenberg, Germany. Air flows are indicated by arrows, the measurement zones are described in the upper part of the schematic.**



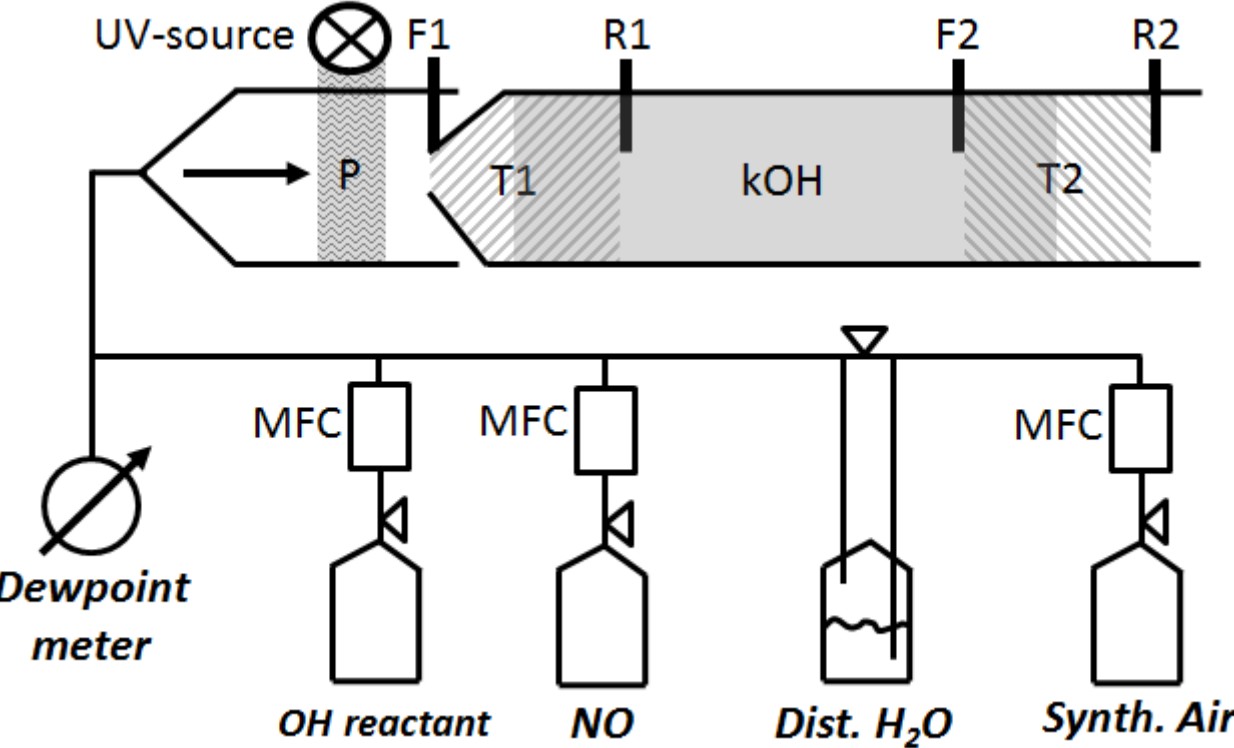

**Figure 2: Schematic of CIMS flow tube with two titration zones: P = OH and HO$_2$ production from H$_2$O photolysis. T1 zone (left hatch) = first titration zone (F1 up to R1) with H$_2$SO$_4$ from OH titration by SO$_2$ injection at F1. T2 zone (right hatch) = second titration zone (F2 up to R2) with H$_2$SO$_4$ from OH titration by SO$_2$ injection at F2. R1 and R2 = added propane to stop H$_2$SO$_4$ production from recycled OH after the first and second titration zone, respectively. kOH (grey solid) = total OH reactivity using ln ([OH]$_{T1}$/[OH]$_{T2}$) indicating [OH]$_{T1,T2}$ is reached after front (F1, F2) and before rear (R1, R2) injection.**



**Figure 3: OH reactivity calibration curve and estimation of scaling rate by linear regression including uncertainties in both measured CIMS ln([OH]$_{T1}$/[OH]$_{T2}$) and propane OH reactivity.**



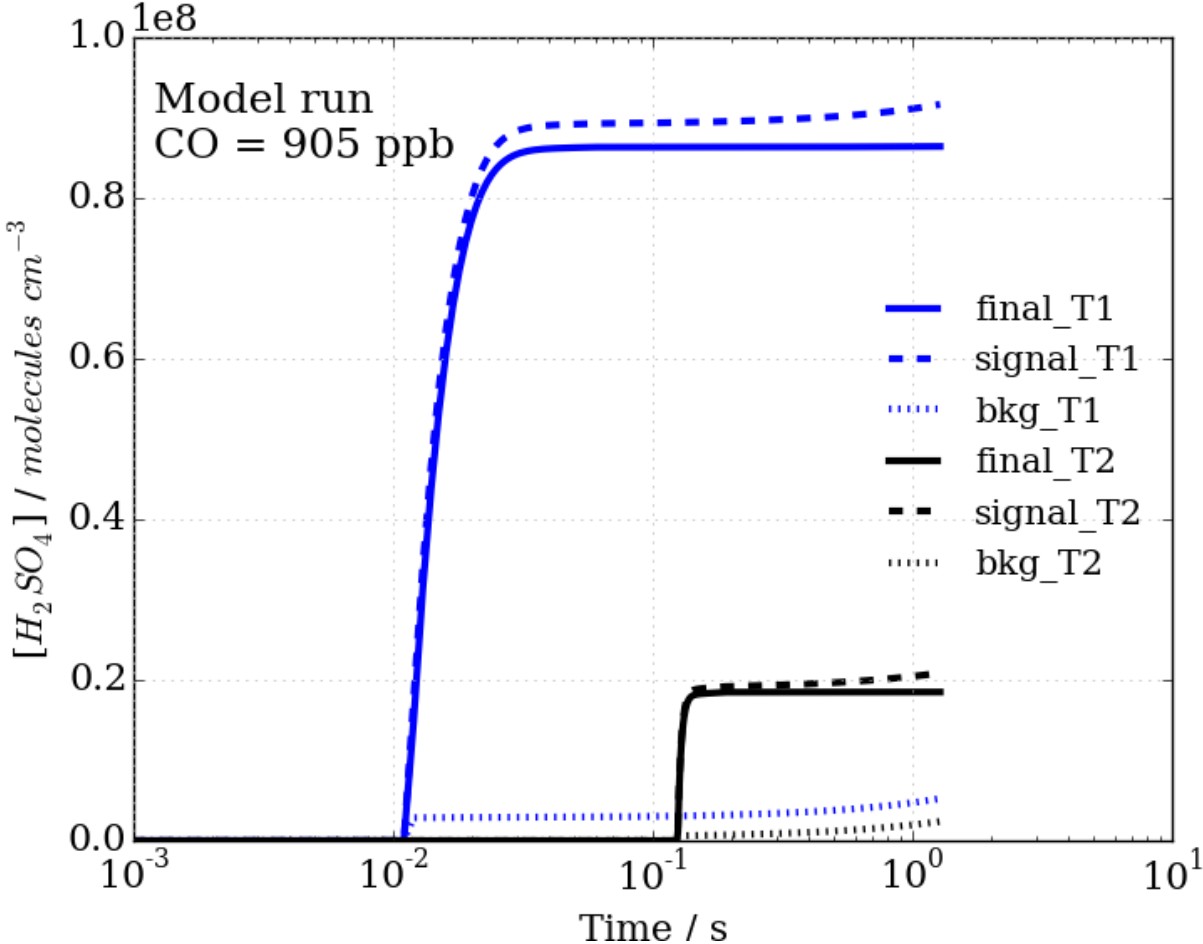

**Figure 4: Sulfuric acid concentrations with time, representing progression in sample flow tube from inlet tip to ionisation region, for signal and background modes (dashed and dotted lines respectively), as well as the calculated difference, leading to final[H₂SO₄]_{T1}= [OH]_{T1} and final[H₂SO₄]_{T2}= [OH]_{T2} concentrations (solid lines) in titration zones 1 (T1) and 2 (T2) which are used to calculate OH reactivity.**





**Figure 5: Variability around mean flowspeed-based calculated scaling rate dependent on 10 min observed pressure and temperature measurements at MOHp in 2015.**



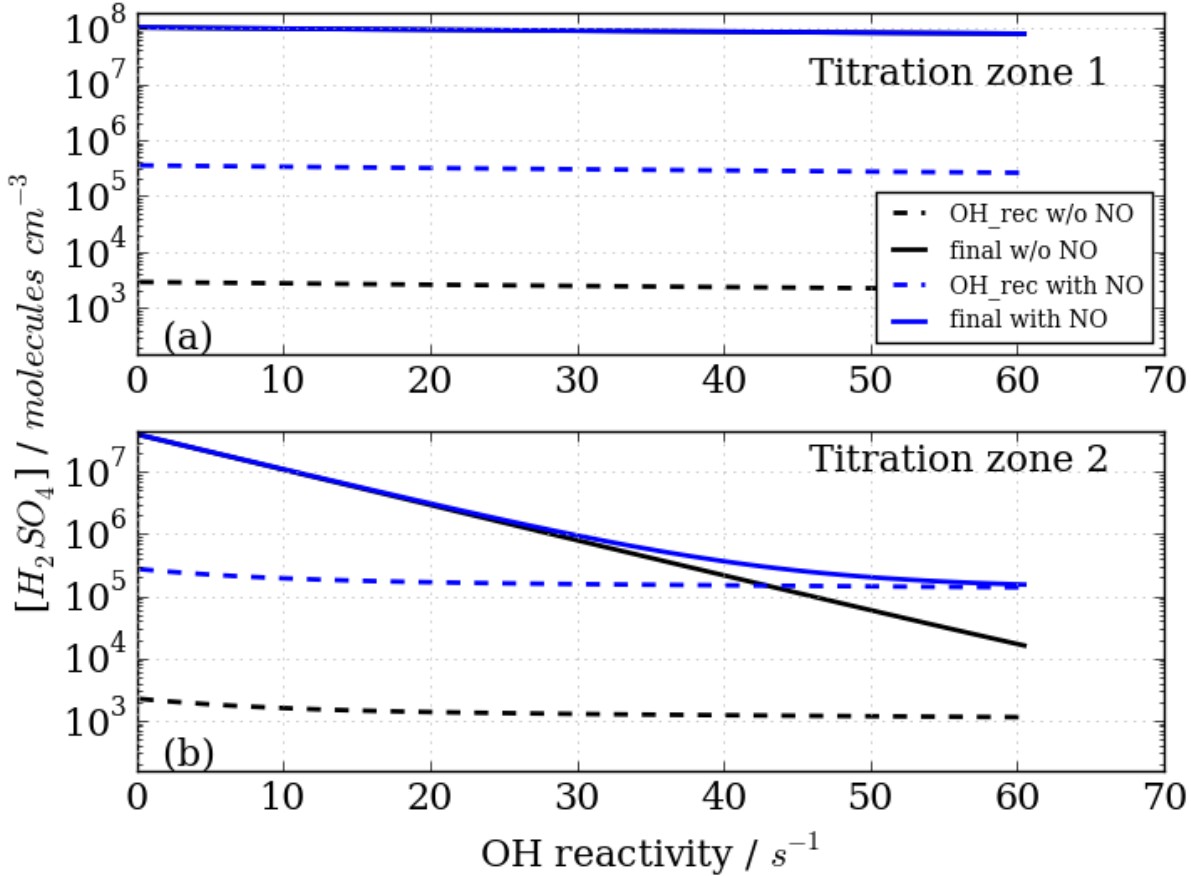

**Figure 6: Modelled CIMS H₂SO₄ concentrations for CO as OH reactant in titration zone 1 (a) and titration zone 2 (b) with and without NO impurity (140 ppt) present. Final concentrations describe differential between total signal and total background mode measurements. OH_rec describe the H₂SO₄ concentrations derived from recycled OH only, contributing to the total measurements.**







**Figure 7:** Modelled CIMS $H_2SO_4$ concentrations and contributions from OH recycling in titration zone 2 with and without NO impurity (140 ppt) present. (a) Absolute $H_2SO_4$ concentrations from recycled OH only in signal and background measurement mode. (b) Total $H_2SO_4$ concentrations. (c) Relative contribution from $OH_{UV}$ and OH recycled to the total (and final) $H_2SO_4$ concentration. (d) Non-linear increases in signal, background and final measurements in case of presence of NO impurity.



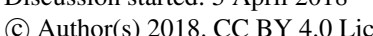

**Figure 8: CIMS OH reactivity underestimation as result of NO-HOx recycling interference in external flow tube NO correction**
5    **experiment. OH reactivity is the average of experiments with OH reactants propane, isoprene, ethene and NO.**





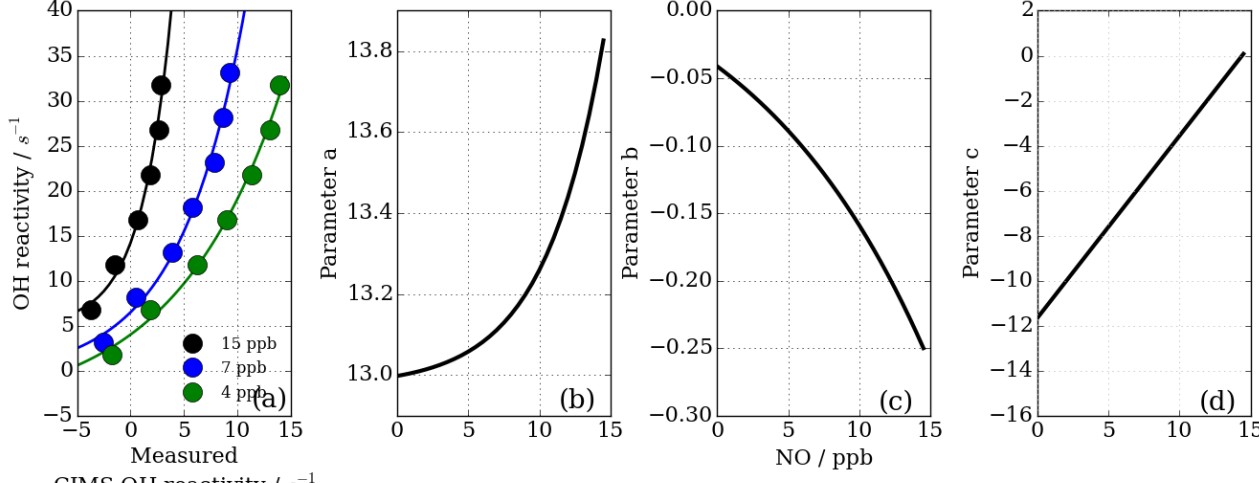

**Figure 9: NO HOx recycling interference on CIMS OH reactivity both dependent on OH reactivity and NO concentration (panel a). Functions of parameters (Eq. 7) are dependent on NO concentration (panels b, c, d correspond to Eq. 8, 9, 10).**





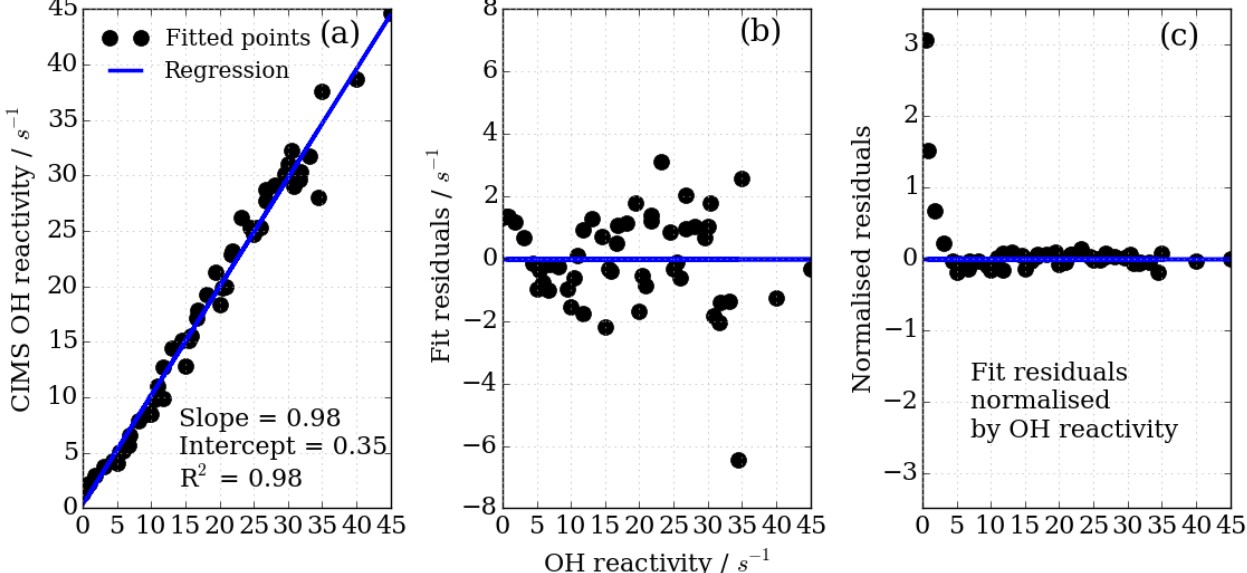

**Figure 10: Performance of the eight parameter correction function (Eq. 7-10), (a) goodness of fit of correction function, (b) fit residuals and (c) fit residuals normalised by OH reactivity.**





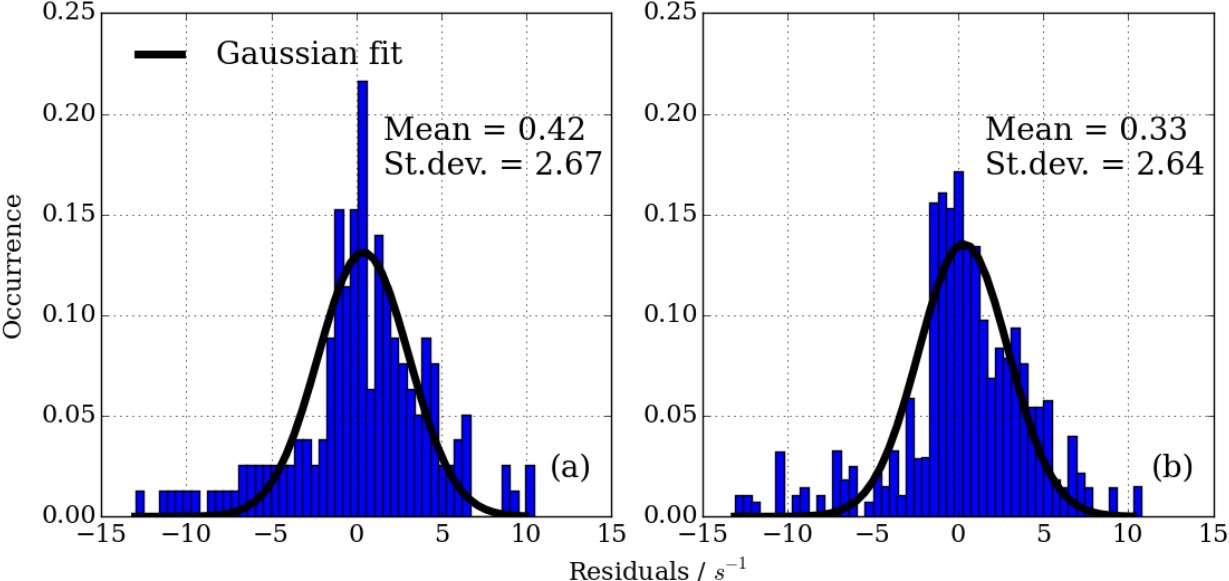

**Figure 11: Residuals in CIMS OH reactivity after applying NO correction function. (a) 165 individual points from isoprene, ethene and propane NO correction experiment, (b) 194 individual points from four propane NO correction experiments March 2016 – March 2017. Distribution is normalised according to number of points in each experiment.**



**Figure 12: OH reactivity underestimation for MOHp relevant NO mixing ratio and typical OH reactivity (maximum 15 s⁻¹). Filled circles are data used for linear regression to obtain mean underestimation of OH reactivity for NO up to 4 ppb.**



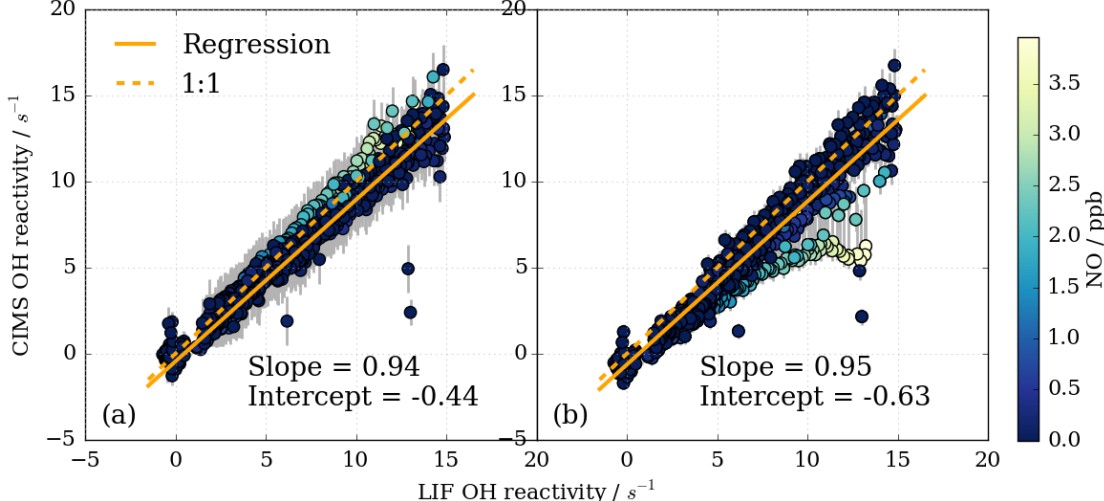

**Figure 13: Subset of OH reactivity measurements from instrument comparison (Fuchs et al., 2017) for OH reactivity up to 15 s$^{-1}$ and NO concentrations up to 4 ppb (716 data points). (a) CIMS data treatment includes eight parameter NO correction, (b) CIMS data are not corrected for NO presence, but a positive NO-dependent uncertainty is associated with the measurement (Fig. 12).**





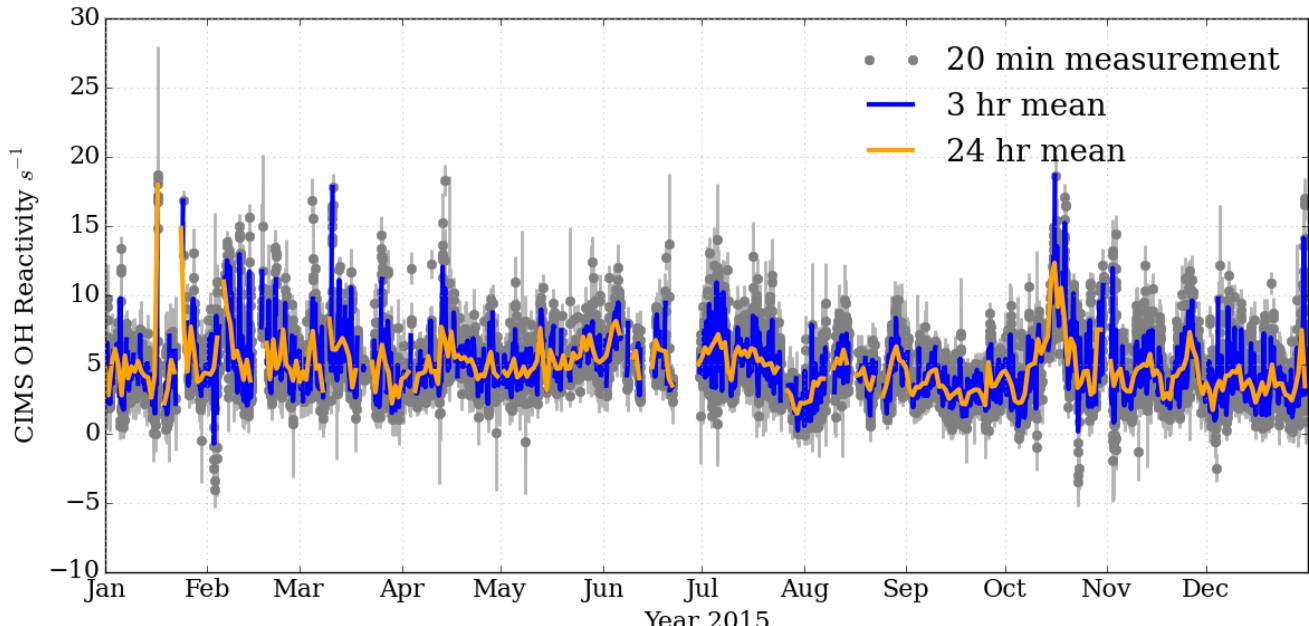

**Figure 14: OH Reactivity measurements by CIMS at MOHp for the year 2015, with 3 hour and daily means in blue and orange respectively. Filtered for recorded instrumental and laboratory interferences and seven outliers not coinciding with short term CO and NO$_2$ concentration peaks. 15541 valid measurement points.**