# Peer review of "A novel semi-direct method to measure OH reactivity by chemical ionisation mass spectrometry (CIMS)"

_Atmospheric Measurement Techniques, 2018_

## Referee Comment (RC1) · Anonymous Referee #1 · 15 May 2018

A novel semi-direct method to measure OH reactivity by chemical ionisation mass spectrometry (CIMS)

Jennifer B.A. Muller et al., Atmos. Meas. Tech. Discuss., doi: 10.5194\amt-2018-99

The authors present the development, characterisation and implementation of a novel instrument to make long-term measurements of OH reactivity (kOH) at the Global Atmosphere Watch (GAW) site at Hohenpeissenberg, Germany.

OH reactivity is a key measurement in atmospheric science, representing the total OH loss rate and used to provide valuable information regarding the oxidising capacity of the atmosphere. However, long-term measurements of such a key contribution to our

understanding of atmospheric composition are currently lacking owing to the complexity of the majority of techniques used to measure this parameter. This work makes therefore makes a significant contribution to the field by describing the adaptation of an existing technique to make long-term measurements.

The paper is generally well-written, and the instrument and its characterisation are well described. While there are significant limitations to the application of the technique to certain environments, these limitations are discussed in detail, with supporting model calculations to provide confidence in the intended application of the technique for long-term measurements in a relatively clean location. I therefore recommend publication after minor comments (listed below) have been addressed.

Page 1, line 13: Please consider re-phrasing 'between below 1 and 40', perhaps to 'from less than 1 to 40'.

Page 1, line 27: 'termed also' to 'also termed'.

Page 3, line 17: 'produced O(1D)' to 'O(1D) produced' (also subscript in CF2 above).

Page 6, Section 3: Is there any potential for photolysis of carbonyl compounds? What would the impact be?

Page 8, line 12: Remove 'such' in 'such e.g. OH. . .'.

Page 9, line 26: Change to 'a range of VOCs and calculated OH reactivity'.

Page 9, line 29: Can you comment on how long 'extensive flushing' is required for?

Page 10: Is it possible for future experiments to purify the SO2 used?

Page 12, line 28: 'from reaction chain' to 'from the reaction chain' (both instances).

Page 13, line 3: Is this the rate or the rate coefficient?

Page 14, line 17: 'inorganic species'.

Page 15, line 20: 'systemacity in errors' to 'systematic errors'.

Page 17, line 3: 'fit parameter' to 'fit parameters'.

Page 17, line 12: Please re-phrase 'highly performant'.

Page 19, line 10: '. . .observed to have a clear diurnal. . .'.

Page 19, line 12: '. . .dominance of biogenic VOCs in summer is contrasted by higher concentrations. . .'.

Page 19, line 31: 'a representative ambient air sample'.

---

## Referee Comment (RC2) · Anonymous Referee #2 · 15 May 2018

The manuscript by Muller et al. described a new method for OH reactivity measurement by CIMS technique, which can only be realized by LIF-Fage technique before. The authors introduced the instrument setup as well as the calibration method and discussed the sources of uncertainty in great detail. The interference from HOx recycling by NO, known as the major uncertainty when measuring OH reactivity, was explored by box model simulations and laboratory experiments. Empirical equations were used to correct the underestimation of OH reactivity by NO interference under different situations and proved to be applicable. The chamber inter-comparison measurements shown the ability of the instrument to take measurements in 1~15 s-1 even with several ppb of NO. Finally the 1-year ambient measurement results at MOPh was outstanding and

are helpful for better understanding the tropospheric oxidative capacity. The restriction of application in low OH reactivity and low NO environment is also clearly stated. The manuscript is well written and I would recommend for publication after the authors addressed the following comments:

Line 31 on Page 3: With 13 SLM sample flow through the 0.019 m diameter chemical reactor sample tube. . .

Comment: The flow rate control in the sample tube is critical in the instrument setup since it is directly related to the scaling rate as well as wall loss rate. It is not clearly stated in the manuscript what the flow rate, how the flow is controlled and what the size of the sample tube in Fig.1 is. Is the pressure in the sample tube close to ambient pressure? All these information will be helpful to understand the reactions between two titration zones.

Line 6 on Page 7: Additionally, changes in volume flow rates can also affect flow characteristics in the sample tube, and therefore OH wall loss and. . .

Line 23 on Page 9: However the wall loss rate can be affected as the measurement of zero reactivity is done with synthetic air . . .

Comment: The wall loss rate is a common issue of OH reactivity measurement and can be caused by various reasons. the wall loss rate can be measured by zero reactivity measurements as well as OH reactivity calibrations, which shall give similar results. In the manuscript the causes of the wall loss rate were discussed in several parts but little data were shown. A more detailed the wall loss rate measurement results will be helpful to prove the stability and capability of the instrument.

Line 18 on Page 14: The effect of OH reactivity underestimation was considerably stronger for CO than for the other OH reactants, so only data from propane, ethane and isoprene experiments were used to derive a correction function. . .

Comment: In Page 19 Line 14, the authors mentioned 'inorganic trace gases such as

[Figure]

CO and NO2 which contribute greatly to the total OH reactivity at MOPh'. Why was the CO experiments not used when CO is the main contributor of OH reactivity at MOPh?

Minor comments:

Line 17 on Page 14: 'specie' should be 'species'

Line 19 on Page 14: 'ethane' should be 'ethene'

---

## Referee Comment (RC3) · Anonymous Referee #3 · 17 May 2018

This manuscript reports a methodology to measure total OH reactivity using Chemical Ionization Mass Spectrometry (CIMS) instruments initially designed to monitor ambient OH radicals. This methodology was implemented on the Hohenpeissenberg CIMS instrument, which can now sequentially measure OH, sulfuric acid, and total OH reactivity. The ability of performing long-term measurements of total OH reactivity, together with the OH radical, is of particular interest for the scientific community to assess long-term trends in the ambient load of OH co-reactants and in the atmospheric oxidation capacity.

The methodology is well described, the performances of the CIMS setup well pre-

sented, and the authors have carefully documented its current limitations. 0-D modeling results are very useful to investigate and quantify these limitations. It is shown that this technique is suitable for the Hohenpeissenberg area, and other areas exhibiting low NO mixing ratios.

This manuscript is clear and concise, and I therefore recommend publication in AMT after the authors address the following comments:

1/ Limitations and uncertainties associated to this measurement technique are thoroughly investigated in section 3. This is a relatively long section and it would be beneficial for the reader to insert an additional table at the end this section to summarize all the sources of uncertainties.

2/ Was the photolysis of ambient OVOCs by the mercury lamp radiation investigated? The amount of radicals produced from OVOC photolysis will not likely be significant compared to the HOx production from water photolysis but could this lead to a significant change in kOH?

Minor comments:

P3 L11: "The CIMS system is frequently calibrated...". Please indicate how often calibrations are performed.

P3 L14: How is the generated OH concentration determined during calibration experiments?

P4 L17-18: "...leading to the quantification of the contribution from recycling and other artifacts...". The term "recycling" may not be the most appropriate here since it does not include OH radicals produced from HO2+NO in the CIMS inlet in the presence of SO2 and propane. I would not use this term.

P5 L16: How was the reaction time inferred? Please clarify whether this is calculated or measured?

P5 L24: Please report values for the generated air flow rate and the CIMS sampling flow rate. The CIMS sampling flow rate of 2280 mL/min is reported P11 but this is too late.

P8 L20: The authors should explain (or remind the reader) why the OH reactivity value is integrated over 10 min but the time resolution is twice longer (20 min). We would expect a time resolution of 10 min.

P10 L12-13: How did the authors come up with NO contamination levels of 140 ppt and 20 ppt in the SO2 and zero air mixtures, respectively? Was it measured? Or are these conservative upper limits? Something else?

P11 L5: Please double-check the uncertainty values stated for the scaling rate. How were they calculated? The $1\sigma$ determination using the IUPAC rate constant for OH+propane seems very small. The IUPAC compilation indicates that $\Delta$logk for OH+propane is $\pm0.08$ at 298K, which I think translates into a $1\sigma$ uncertainty of approximately 10%. How can the uncertainty on the scaling rate be lower than the uncertainty on the rate constant?

P11 section 3.3: This section compares scaling rate values derived from flow tube calibrations to values inferred from atmospheric chamber experiments. This provides a nice validation of the flow tube calibration method. The authors should also discuss the comparison of kw values derived from these different experiments? Were the kw values consistent within uncertainty?

P12 L14-15: How is the upper limit for OH reactivity measurements quantified? Is the calculation based on a $3\sigma$ detection limit of OH in zone 2?

P12 L26: Sup. mat. S1 is cited for more details about the tagged mechanism. However there is no information about the tagging feature in S1.

P14 Eqs. 7-10: How was this complex empirical function determined? Can physical meanings be associated to some of the terms?

P16 L22: A scatter of 10-20% is reported. Under which conditions was it observed (nature of trace gases, OH reactivity level, etc.). How many $\sigma$ is it?

P17 L7-9: Please be quantitative. How accurate is "high accuracy" and "lower accuracy"? Table 1: How many $\sigma$ did the authors used to calculate the limits of detection? What are the precisions on $\Delta t$ and kw values?

Fig. 14: It would be useful to add a time series for NO since it can disturb the measurements. Could the measurement underestimation due to the presence of NO also be quantified and shown as another time series?

Technical corrections

P1 L27: "...termed also total OH loss rate..." should read "...also termed total OH loss rate..."

P4 L20-23: This sentence should be rephrased. It's not clear what is meant by "... is the final OH concentration without UV light subtracted from the final OH concentration with the UV lamp on."

P4 Eq. 1: Missing negative sign in the exponential factor. kw should also be added in this equation for consistency with Eqs. 2-4

P4 L32: Delete the word "lawful"

P6 L7: "... within the flow tube from point of the inlet to ..." should read "... within the flow tube from the inlet tip to ..."

P6 L18-19: "The scaling rate, i.e. the times of SO2/propane injection...". Shouldn't it read "The scaling rate, the inverse of the reaction time between T1 and T2..."

P7 L3-6: "volume flow" should read "volumetric flow rate". Several instances.

P7 L12: "Additionally" should read "Additional"

P9 L26: "synthetic air has been measured for a range of VOC and OH reactivity calculated" should read "synthetic air has been screened for a range of VOC and OH reactivity calculated from a quantification of these contaminants"

P13 L18: Replace "annulled" by "cancelled"

P19 L24: "Calculations and propagation of uncertainties was performed..." should read "Calculations and propagation of uncertainties were performed..."

P19 L27: Double-check the 5% value reported for LN([OH]T1/[OH]T2). It's 7.1% in section 3.2.1.

P19 L31-32. Some units are missing.
* * *

---

## Author Comment (AC1) · 27 Jun 2018

***Author's response to the review of***

"A novel semi-direct method to measure OH reactivity by chemical ionisation mass spectrometry (CIMS)" by Jennifer B.A. Muller et al., Atmos. Meas. Tech. Discuss., doi: 10.5194\amt-2018-99

The authors greatly appreciate the comments, suggestions and follow-on questions provided by all three anonymous referees. All points made by the referees are addressed in the following sections.

**Referee #1**

Page 1, line 13: Please consider re-phrasing 'between below 1 and 40', perhaps to 'from less than 1 to 40'.

*Response: Re-phrased as suggested.*

Page 1, line 27: 'termed also' to 'also termed'.

*Response: Changed.*

Page 3, line 17: 'produced O(1D)' to 'O(1D) produced' (also subscript in CF2 above).

*Response: Changed and subscript in "$CaF_2$" corrected.*

Page 6, Section 3: Is there any potential for photolysis of carbonyl compounds? What would the impact be?

*Response: We assume the referee refers to measurements in ambient air. Also, as no carbonyl compounds have been used for calibration, this response focusses on ambient air only.*

*Yes, in principle there is potential for photolysis of carbonyl compounds within the UV zone. Using atmospherically common aldehydes (formaldehyde, acetaldehyde, propionaldehyde) as an example, the impact would be that the photolysis can lead to some products (here H) which have a lower OH reactivity than the aldehyde itself. However, some products have larger rate constants (e.g. HCO + OH). Reaction rate constants of OH with the products can be different by a few orders of magnitude than those with aldehydes, see Table below, rate constants for k(298K) from NIST Chemical Kinetics Database, IUPAC and JPL Recommendations.*

*Response-Table 1: Reaction Rate constants OH + aldehydes and OH + aldehyde photolysis products (only major product channels considered here).*

| Units: cm$^3$ molecules$^{-1}$ s$^{-1}$ | Formaldehyde HCHO | Acetaldehyde CH$_3$CHO | Propionaldehyde C$_2$H$_5$CHO |
|---|---|---|---|
| *Reaction with carbonyl* | $k_{OH+HCHO}$ = 9.38 x 10$^{-12}$ | $k_{OH+CH3CHO}$ = 1.5 x 10$^{-11}$ | $k_{OH+C2H5CHO}$ = 2 x 10$^{-11}$ |
| *Reaction with photolysis product 1* | $k_{OH+H}$ = 9.9 x 10$^{-17}$ *(products H$_2$ +O)* | $k_{OH+CH3}$ = 8 x 10$^{-11}$ | $k_{OH+C2H5}$ = 4 x 10$^{-11}$ *(products C$_2$H$_4$ +H$_2$O)* |
| *Reaction with photolysis product 2* | $k_{OH+HCO}$ = 1.83 x 10$^{-10}$ | $k_{OH+ HCO}$ = 1.83 x 10$^{-10}$ | $k_{OH+HCO}$ = 1.83 x 10$^{-10}$ |

*Although fast, the reaction of OH with these products of photolysis will be less important than their reaction with O$_2$. The reaction with oxygen to produce OH, HO$_2$ or RO$_2$ radicals is often almost equally fast: the reaction H + O$_2$ → OH + O is fast with k(298K) ~ 3.5 x 10$^{-11}$ cm$^3$ molecules$^{-1}$ s$^{-1}$ and the reaction H + O$_2$ + M → HO$_2$ + M similarly so at k(298K)~ 1.2 x 10$^{-12}$ cm$^3$ molecules$^{-1}$ s$^{-1}$ for [M$_{MOHP}$] = 2.3 x 10$^{19}$ molecules cm$^{-3}$. For HCO + O$_2$ → CO +HO$_2$ , k(298K) is ~5.1 x 10$^{-12}$ cm$^3$ molecules$^{-1}$ s$^{-1}$. For CH$_3$ + O$_2$ +M → CH$_3$O$_2$ + M, k(298K) is ~8 x 10$^{-13}$ cm$^3$ molecules$^{-1}$ s$^{-1}$. For C$_2$H$_5$ + O$_2$ → C$_2$H$_4$ +HO$_2$ , k(298K) is ~3.8 x 10$^{-15}$ cm$^3$ molecules$^{-1}$ s$^{-1}$.*

*If dissociation by photolysis was to significantly reduce the aldehyde concentration, this could lead to an underestimation of measured OH reactivity. However back-of-the-envelope calculations show that only a fraction of a percent of carbonyl compounds photolyses in the CIMS UV zone at λ = 185 nm (Response-Table 2 below). This is a negligible amount and therefore can be ignored as a source of uncertainty for the OH reactivity measurements.*

*Response-Table 2: Back-of-the-envelope calculations of aldehyde loss by photolysis in CIMS UV zone. Assumptions and system conditions also listed.*

| | Formaldehyde HCHO | Acetaldehyde CH$_3$CHO | Propionaldehyde C$_2$H$_5$CHO |
|---|---|---|---|
| *†Absorption cross section σ$_\lambda$ at λ ~185 nm [cm$^2$ molecule$^{-1}$]* | 1.15 x 10$^{-18}$ | 3 x 10$^{-20}$ | 1 x 10$^{-17}$ |
| *Quantum yield q$_\lambda$ assumed (upper limit) [molecules photon$^{-1}$]* | 1 | 1 | 1 |
| *Photon flux I [photons cm$^{-2}$s$^{-1}$]* | 4 x 10$^{12}$ | 4 x 10$^{12}$ | 4 x 10$^{12}$ |
| *Photolysis rate constant k = q$_\lambda$ σ$_\lambda$ I [s$^{-1}$]* | 4.6 x 10$^{-6}$ | 1.2 x 10$^{-7}$ | 4 x 10$^{-5}$ |
| *Typical mixing ratio at MOHp [ppb]* | 2 | 1.4 | 0.4 |
| *Concentration at MOHp [molecules cm$^{-3}$]* | 4.6 x 10$^{10}$ | 3.2 x 10$^{10}$ | 9.2 x 10$^9$ |
| *Residence time in UV zone [s]* | 0.0034 | 0.0034 | 0.0034 |
| *% of carbonyl photolysed* | 1.6 x 10$^{-4}$ | 4 x 10$^{-8}$ | 1.4 x 10$^{-5}$ |
| *Change in OH reactivity* | 7 x 10$^{-5}$ | 2 x 10$^{-8}$ | 3 x 10$^{-6}$ |

| | | | |
|---|---|---|---|
| $[s^{-1}]$ | | | |

† *Mainz UV-VIS spectral atlas* http://satellite.mpic.de/spectral_atlas *; Keller-Rudek, H., Moortgat, G. K., Sander, R., and Sörensen, R.: The MPI-Mainz UV/VIS spectral atlas of gaseous molecules of atmospheric interest, Earth Syst. Sci. Data, 5, 365–373, (2013), DOI: 10.5194/essd-5-365-2013*

*A sentence to address this comment, and the comment of referee #3 regarding OVOC photolysis (see later), has been added on page 6, introductory paragraph Section 3:*

*"Effects from photolysis of organic OH reactants such as e.g. aldehydes and alcohols in the CIMS UV zone have been estimated to be negligible and are not discussed further in this section."*

Page 8, line 12: Remove 'such' in 'such e.g. OH. . .'.

*Response: Removed.*

Page 9, line 26: Change to 'a range of VOCs and calculated OH reactivity'.

*Response: Changed.*

Page 9, line 29: Can you comment on how long 'extensive flushing' is required for?

*Response: 'Extensive flushing' is admittedly a vague description. In our case, both flushing of the lines with 100 ml/min for one hour, as well as flushing with a large flow (several L/min) for several minutes was sufficient to remove trace contamination. However what constitutes sufficient flushing depends on the set up and length of tubing. To clarify, the phrase 'extensive flushing' has been changed and the sentence now reads: "It was observed that tubing, valves etc. needed to be sufficiently flushed as to remove trace contamination and achieve a low constant level of residues before carrying out the zero OH reactivity measurements."*

Page 10: Is it possible for future experiments to purify the $SO_2$ used?

*Response: Purification and obtaining a $SO_2$ gas mixture without traces of NO is something we are currently working on. First tests on commercial mixtures of $SO_2$ indicate that $SO_2$ in synthetic air contain lower level of NO contamination than $SO_2$ in $N_2$. We are continuing to monitor the trace levels of NO in all of the $SO_2$ mixtures we use and remain in contact with the compressed gas suppliers to understand the source of NO contamination. We are also planning to test set ups to chemically remove NO by adding ozone to the $SO_2$ gas stream prior to the injection into the CIMS system. The set up will aim to remove the ozone after the reaction with NO has completed. The effect of injecting the reaction product $NO_2$ into the CIMS has been estimated to be negligible for OH reactivity measurements. Although the reaction rate constant of $NO_2$ + OH is one order of magnitude greater (k $\sim 10^{-11}$ cm$^3$ molecules$^{-1}$ s$^{-1}$) than for $SO_2$ + OH (k$\sim 10^{-12}$ cm$^3$ molecules$^{-1}$ s$^{-1}$), the $SO_2$ mixing ratio (~10*

*ppm) in the sample flow is six orders of magnitudes larger than the $NO_2$ mixing ratio (~100 ppt). For zero OH reactivity measurements, the produced $NO_2$ is estimated to make up less than 0.05 $s^{-1}$.*

Page 12, line 28: 'from reaction chain' to 'from the reaction chain' (both instances).

*Response: Changed, "the" added.*

Page 13, line 3: Is this the rate or the rate coefficient?

*Response: This is the rate coefficient; "rate" has now been unambiguously defined and changed to "reaction rate constant"*

Page 14, line 17: 'inorganic species'.

*Response: Changed.*

Page 15, line 20: 'systemacity in errors' to 'systematic errors'.

*Response: Changed.*

Page 17, line 3: 'fit parameter' to 'fit parameters'.

*Response: Changed, "s" added.*

Page 17, line 12: Please re-phrase 'highly performant'.

*Response: Wording was changed from "The FZJ LIF instrument was shown to be highly performant throughout the campaign" to "The FZJ LIF instrument performed very well throughout the campaign"*

Page 19, line 10: '. . .observed to have a clear diurnal. . .'.

*Response: Changed, article "a" inserted.*

Page 19, line 12: '. . .dominance of biogenic VOCs in summer is contrasted by higher concentrations..'

*Response: Changed.*

Page 19, line 31: 'a representative ambient air sample'.

*Response: Changed.*

(End Referee #1)

**Referee #2**

Line 31 on Page 3: With 13 SLM sample flow through the 0.019 m diameter chemical reactor sample tube. . .

Comment: The flow rate control in the sample tube is critical in the instrument setup since it is directly related to the scaling rate as well as wall loss rate. It is not clearly stated in the manuscript what the flow rate, how the flow is controlled and what the size of the sample tube in Fig.1 is. Is the pressure in the sample tube close to ambient pressure? All these information will be helpful to understand the reactions between two titration zones.

*Response: The sample flow is around 13 SLM and the dimensions of the sample tube are: diameter of 0.019 m and length of 0.30 m. The complete length from sample tube tip to the pinhole is 0.8 m. The pressure in the sample tube is close to ambient pressure (around 900 hPa at MOHp). The sample flow of 13 SLM is not itself kept constant by a mass flow controller but is the result of the difference of several other mass flow controlled flows in the system: Addition of 84 SCCM per injector (i.e. total of 168 SCCM), addition of sheath air is maintained at 37.0 SLM, $HNO_3$ enriched flow is kept at 5.0 SCCM, addition of pinhole synthetic air is controlled at 1.1 SLM and the total flow that is removed from the ionisation region is maintained at 51.0 SLM. The resulting balance of flows (51.0 – 0.168 - 37.0 - 1.1 - 0.005 = 12.727 SLM) equals the sample flow. In order to obtain the residence time between the titration zones (F1 to F2 = 0.15 m) the volumetric flow rate has to be calculated and the mass flow rates have to be corrected for the ambient temperature and pressure.*

*Salient points on the sample flow tube and the flow set up has been added in the manuscript: "The CIMS sample flow tube dimensions are 0.019 m in diameter and 0.30 m in length, with a sample flow of around 13 SLM. The complete length from sample tube tip to the pinhole is 0.8 m. The distance between the front injectors is 0.15 m and between front and rear injectors is 0.054 m (Fig. 1). The pressure in the sample tube is close to ambient pressure (around 900 hPa at MOHp). The sample flow of 13 SLM is not itself kept constant by a mass flow controller but maintained by the difference of other mass flow controlled flows. The total residence time within the flow tube is circa 0.9 s which is long enough for HOx recycling to become significant inside the flow tube and the measurement artefacts from recycling thus need to be considered and accounted for."*

Line 6 on Page 7: Additionally, changes in volume flow rates can also affect flow characteristics in the sample tube, and therefore OH wall loss and. . .

Line 23 on Page 9: However the wall loss rate can be affected as the measurement of zero reactivity is done with synthetic air . . .

Comment: The wall loss rate is a common issue of OH reactivity measurement and can be caused by various reasons. The wall loss rate can be measured by zero reactivity measurements as well as OH reactivity calibrations, which shall give similar results.

In the manuscript the causes of the wall loss rate were discussed in several parts but little data were shown. A more detailed the wall loss rate measurement results will be helpful to prove the stability and capability of the instrument.

*Response: We agree that the determination of wall loss rates represent a key aspect of any OH reactivity measurement system. Wall loss rates $k_w$ have now been included in Table 2, showing the validation of wall loss rates determined with the experimental set up using the external glass flow tube are equivalent to the measurements in an operational set up (FZJ SAPHIR chamber, Fuchs et al., 2017).*

*To further illustrate the quality of the wall loss measurements, zero OH reactivity was measured every day during the OH reactivity comparison campaign in April 2016 at the SAPHIR chamber. At the beginning of each day, the chamber was humidified without addition of OH reactants, i.e. zero OH reactivity in the chamber, so consequently the wall loss rates were measured for 0.5-1 hour. The time series of these measurements are shown here:*

[Figure]

*The wall loss rates are overall stable for each individual day, and also for the whole campaign period from 7-15/4/2016.*

*Looking at stability over longer periods of time (half a year), the CIMS wall loss rates in Table 2 give indication of a similar level of variability and uncertainty compared to the shorter term stability shown above.*

*The above response has been added to the Supplementary material as S3 and is referenced in Section 3.3 "Validation of CIMS OH reactivity calibration using external glass flow tube".*

Line 18 on Page 14: The effect of OH reactivity underestimation was considerably stronger for CO than for the other OH reactants, so only data from propane, ethane and isoprene experiments were used to derive a correction function. . .

Comment: In Page 19 Line 14, the authors mentioned 'inorganic trace gases such as CO and NO2 which contribute greatly to the total OH reactivity at MOPh'. Why was the CO experiments not used when CO is the main contributor of OH reactivity at MOPh?

*Response: Considering long-term medians, CO alone makes up typically 15 - 25 % of the total measured OH reactivity. When using only CO as OH reactant for the NO response experiments, CO*

*mixing ratios of up to about 10 ppm have to be used. Since $HO_2$ is directly produced as OH reacts with CO, the non-linear recycling process of $HO_2$ + NO = OH is strongly amplified which is not comparable with the effect expected during ambient air measurements. Therefore the results from CO and NO calibration experiments are not directly applicable and transferrable to ambient air measurements. However for e.g. propane the NO calibration experiments can be applied to ambient measurements as propane + OH does not produce this non-linear amplification of recycling.*

*To clarify, the sentence was rephrased as follows: "The effect of OH reactivity underestimation was considerably stronger for CO than for the other OH reactants, because NO induced OH recycling is amplified when OH reacts with CO to additionally produce $HO_2$ in the CIMS system. These non-linear results from the external glass flow tube experiments with large $HO_2$ production are therefore not directly applicable and transferrable to ambient air measurements. Thus only data from propane, ethene and isoprene experiments were used to derive a correction function that would be representative for an ambient air matrix."*

Minor comments:

Line 17 on Page 14: 'specie' should be 'species'

*Response: Changed.*

Line 19 on Page 14: 'ethane' should be 'ethene'

*Response: Changed.*

(End Referee #2)

**Referee #3**

1/ Limitations and uncertainties associated to this measurement technique are thoroughly investigated in section 3. This is a relatively long section and it would be beneficial for the reader to insert an additional table at the end this section to summarize all the sources of uncertainties.

*Response: An overview table has been put together and as suggested included in Section 3. Reference to the table is made in the introductory paragraph on page 7. Consequently, Table 2 has become Table 3, and the overview table below is Table 2. All references to previous "Table 2" have been changed to "Table 3".*

*Reference to Table 2 reads: "All uncertainties, limitations and their relevance to the scaling rate, the wall loss rate and ambient measurements discussed in this section are shown in Table 2."*

*Table 2: Overview of uncertainties (1σ) and limitations discussed in Section 3, indicating relevance to scaling rate $sr_{CIMS}$ determination, wall loss rate $k_w$ determination and/or relevance for ambient measurements.*

| Section # | Source of uncertainty | Relevant for $sr_{CIMS}$ determination | Relevant for $k_w$ determination | Relevant for ambient measurements |
|---|---|---|---|---|
| 3.1 | Changes in ambient pressure and temperature | | | Uncertainty around mean ± 0.17 $s^{-1}$

 Maximum systematic deviation ± 0.4 $s^{-1}$ |
| 3.2.1 | Measurement of OH by CIMS | | $[OH]_{T1}$ median variability 1.5 %
 $[OH]_{T2}$ median variability 2.7 % | $ln([OH]_{T1}/[OH]_{T2})$ typical uncertainty 7.1 % |
| 3.2.2 | OH reactant concentration in calibration gas mixture | Uncertainty in 1 % CO gas mixture = 2 % in OH reactivity
 Uncertainty in 0.2 % propane gas mixture = 5 % in OH reactivity | | |
| 3.2.3 | OH reactant contamination in calibration gas mixture | Not detectable here, see text. | | |
| 3.2.4 | OH reactant contamination in carrier gas (here synthetic air) | | 0.02 ± 0.02 $s^{-1}$
 Contamination level can vary, see text. | |
| 3.2.5 | NO contamination in all gas mixtures | For 0-40 $s^{-1}$, $sr_{CIMS}$ overestimation 5 %
 For 20-40 $s^{-1}$, $sr_{CIMS}$ overestimation 9 %
 For 0-20 $s^{-1}$, $sr_{CIMS}$ overestimation 1 % | | NO up to 380 ppt in sample flow from $SO_2$ gas mixture |
| 3.2.6 | OH kinetic rate constants | Uncertainty in $sr_{CIMS}$ for CO 11 %
 Uncertainty in $sr_{CIMS}$ for propane 5 % | | |
| 3.3 | Calibration using external glass flow tube | Jülich mean 9.1 ± 0.4 $s^{-1}$
 MOHp mean 9.7 ± 0.5 $s^{-1}$ | Jülich mean 8.6 ± 0.5 $s^{-1}$
 MOHp mean 9.4 ± 0.8 $s^{-1}$ | |
| 3.4 | Upper measurement limit | 40 $s^{-1}$ | | 40 $s^{-1}$ |
| 3.5 | Ambient NO leading to instrument internal HOx recycling | NO up to 15 ppb, non-linear function, see text | | NO up to 4 ppb, underestimation 0.8 $s^{-1}$/ppb |

2/ Was the photolysis of ambient OVOCs by the mercury lamp radiation investigated? The amount of radicals produced from OVOC photolysis will not likely be significant compared to the HOx production from water photolysis but could this lead to a significant change in kOH?

*Response:*

*In addition to the response given to the comment raised by Referee #1 about the photolysis of carbonyl compounds, photolysis of methanol ($CH_3OH$) and ethanol ($C_2H_5OH$) are considered here as additional atmospherically common OVOC species. Approach for estimation same as above (response to Referee #1).*

*Response-Table 3. Reaction rate constants for OH with alcohols (OVOC) and with alcohol photolysis products. Only major product channels considered.*

| Units: $cm^3$ molecules$^{-1}$ s$^{-1}$ | Methanol $CH_3OH$ | Ethanol $C_2H_5OH$ |
|---|---|---|
| Reaction with alcohol | $k_{OH+CH3OH} = 9 \times 10^{-13}$ | $k_{OH+C2H5OH} = 3.2 \times 10^{-12}$ |
| Reaction with photolysis product 1 | $k_{OH+H} = 9.9 \times 10^{-17}$ (products $H_2$ +O) | $k_{OH+H2} = 6.7 \times 10^{-15}$ |
| Reaction with photolysis product 2 | $k_{OH+CH3O} = 3 \times 10^{-11}$ | $k_{OH+CH3CHO} = 1.5 \times 11^{-11}$ |
| Reaction with photolysis product 3 | $k_{OH+H2} = 6.7 \times 10^{-15}$ | |
| Reaction with photolysis product 4 | $k_{OH+CH2O} = 4.38 \times 10^{-12}$ | |

In the case where H (or $H_2$) is a product of photolysis, H will actually react faster with oxygen to produce OH or $HO_2$ radicals: the reaction H + $O_2$ → OH + O is fast with k(298K) ~ 3.5 x 10$^{-11}$ cm$^3$ molecules$^{-1}$ s$^{-1}$ and the reaction H + $O_2$ + M → $HO_2$ + M is similarly fast at k(298K)~ 1.2 x 10$^{-12}$ cm$^3$ molecules$^{-1}$ s$^{-1}$ for [$M_{MOHP}$] = 2.3 x 10$^{19}$ molecules cm$^{-3}$.

Also products $CH_3O$ and $CH_2O$ will react with $O_2$ at fast rates (order of 10$^{-15}$ and 10$^{-12}$ cm$^3$ molecules$^{-1}$ s$^{-1}$, NIST).

The photolysis of alcohols leads to products that have greater reaction rate constants with OH than the alcohols themselves (Response-Table 3), so the potential exists for an overestimation of OH reactivity, if the process of photolysis is important enough.

*Response-Table 4. Back-of-the-envelope calculations of alcohol loss by photolysis in CIMS UV zone. Assumptions and system conditions also listed.*

| | Methanol $CH_3OH$ | Ethanol $C_2H_5OH$ |
|---|---|---|
| †Absorption cross section $\sigma_\lambda$ at $\lambda$ ~185 nm [cm$^2$ molecule$^{-1}$] | 6 x 10$^{-19}$ | 1.1 x 10$^{-18}$ |
| Quantum yield $q_\lambda$ assumed (upper limit) [molecules photon$^{-1}$] | 1 | 1 |
| Photon flux I [photons cm$^{-2}$s$^{-1}$] | 4 x 10$^{12}$ | 4 x 10$^{12}$ |
| Photolysis rate constant k = $q_\lambda$ $\sigma_\lambda$ I [s$^{-1}$] | 2.4 x 10$^{-6}$ | 4.4 x 10$^{-6}$ |
| Typical mixing ratio at MOHp [ppb] | 10 | 0.4 |
| Concentration at MOHp | 2.3 x 10$^{11}$ | 9.2 x 10$^9$ |

| | | |
|---|---|---|
| [molecules cm⁻³] | | |
| Residence time in UV zone [s] | 0.0034 | 0.0034 |
| % photolysed | $8 \times 10^{-7}$ | $2 \times 10^{-6}$ |
| Change in OH reactivity [s⁻¹] | $2 \times 10^{-7}$ | $6 \times 10^{-8}$ |

*† Mainz UV-VIS spectral atlas [http://satellite.mpic.de/spectral_atlas](http://satellite.mpic.de/spectral_atlas) ; Keller-Rudek, H., Moortgat, G. K., Sander, R., and Sörensen, R.: The MPI-Mainz UV/VIS spectral atlas of gaseous molecules of atmospheric interest, Earth Syst. Sci. Data, 5, 365–373, (2013), DOI: 10.5194/essd-5-365-2013*

*Response-Table 4 shows that, as for the aldehydes, the photolysis of common OVOC, such as alcohols, in the CIMS UV zone is not sufficient enough to create detectable changes in OH reactivity during ambient air sampling.*

*A sentence to address this comment, and the comment of referee #1 regarding carbonyl compound photolysis (see above), has been added on page 6, introductory paragraph Section 3:*

*"Effects from photolysis of organic OH reactants such as e.g. aldehydes and alcohols in the CIMS UV zone have been estimated to be negligible and are not discussed further in this section."*

Minor comments:

P3 L11: "The CIMS system is frequently calibrated. . .". Please indicate how often calibrations are performed.

*Response: OH concentration calibrations are performed every 20 minutes. This has now been included in the text: "The CIMS system is calibrated every 20 minutes to obtain absolute OH concentrations …" Although calibration frequency has changed since, the method of calibration has remained the same and has been fully described in Berresheim et al. (2000) to which the reader is referred in the text. The following has now also been clarified in the text (page 4, last paragraph of Section 2.1): "It is worth noting that the absolute OH concentrations are not critical for the OH reactivity measurement as the ratio between titration zone 1 and 2 is used (see Section 2.2), i.e. it is a relative measure and the stability of conditions between the two zones is relevant instead. Amongst other things, the development of the CIMS OH reactivity measurement arose out of the synergy of utilising the already existing frequent OH calibration measurement".*

P3 L14: How is the generated OH concentration determined during calibration experiments?

*Response: As mentioned above the absolute OH concentration is not critical for the OH reactivity measurement, therefore we did not include the full description on how the OH concentration is determined during calibration experiments, but rather refer the reader to the previous in-depth report by Berresheim et al.(2000).*

P4 L17-18: ". . .leading to the quantification of the contribution from recycling and other artifacts. . .". The term "recycling" may not be the most appropriate here since it does not include OH radicals

produced from HO2+NO in the CIMS inlet in the presence of SO2 and propane. I would not use this term.

*Response: Point taken. The term "recycling" has been removed here and the sentence now reads: "This involves adding both $SO_2$ and propane at the front injector, leading to the quantification of the contribution from ambient sulfuric acid and artefacts producing sulfuric acid."*

P5 L16: How was the reaction time inferred? Please clarify whether this is calculated or measured?

*Response: This reaction time value is calculated based on the geometry of the CIMS sample tube and the sample flow rate. This has now been clarified in the text: "… the distance between the two titration zones is equivalent to a calculated reaction time of about 0.1 s."*

P5 L24: Please report values for the generated air flow rate and the CIMS sampling flow rate. The CIMS sampling flow rate of 2280 mL/min is reported P11 but this is too late.

*Response: The flow rates through the ambient air inlet, the CIMS sample tube and the OH reactivity calibration glass tube are now specified in this Section 2.3: "Since the inlet air flow rate (Fig. 1) of 2280 L/min is too large to be generated for OH reactivity calibration, an external glass flow tube is placed on top of the sample tube during OH reactivity calibration, schematically shown in Fig. 2. The setup is such that the excess of the OH reactivity calibration flow of 50 SLM can overflow and the normal CIMS sample flow rate is maintained at 13 SLM."*

*A later sentence referring to the external glass flow tube was shortened and adapted as follows: "The OH reactant is delivered in a humidified synthetic air matrix, which is made of Milli-Q ultrapure water, …"*

P8 L20: The authors should explain (or remind the reader) why the OH reactivity value is integrated over 10 min but the time resolution is twice longer (20 min). We would expect a time resolution of 10 min.

*Response: The sentence has been modified and extended to reiterate that OH reactivity is \*one\* of several measurements made sequentially by the CIMS instrument: "Titration zone measurements in zone 2 immediately follow those in zone 1 so that the measured OH reactivity is effectively an average over a ten minute period. However during routine operation the OH reactivity time resolution is actually 20 minutes because the complete operational CIMS measurement sequence includes other separate measurements such as sulfuric acid and peroxyradical concentration measurements."*

P10 L12-13: How did the authors come up with NO contamination levels of 140 ppt and 20 ppt in the $SO_2$ and zero air mixtures, respectively? Was it measured? Or are these conservative upper limits? Something else?

*Response: The NO contamination level of 20 ppt in zero air is based on measurements using a chemiluminescence (CLD) analyser. The measurements showed concentrations of up to a few ppt in zero air, so based on the detection limit of the CLD analyser the value of 20 ppt was used. This constitutes a conservative, yet realistic upper limit.*

*Levels of up to a few hundred ppt NO from the $SO_2$ gas mixtures were confirmed by direct measurements using CLD analysers (as detailed in the same section in the manuscript). However the NO contamination level of 140 ppt in the $SO_2$ mixture is derived from FACSIMILE modelling during the OH reactivity comparison campaign at the Jülich chamber SAPHIR in April 2016 (Fuchs et al., 2017) and constitutes a realistic level of contamination. Here, an effect of OH reactivity suppression (at OH reactivity > 20 $s^{-1}$) was observed during the CO experiments. This effect had not previously observed and by elimination of causes was identified to be the result of the presence of NO in the $SO_2$ gas mixture. Unfortunately direct measurement of NO was not possible at that point anymore since the $SO_2$ gas from this mixture had been expended for routine measurements at MOHp. The CIMS measurement data from the SAPHIR CO chamber experiments were compared to FACSIMILE modelling results for a range of NO contamination levels and so the NO mixing ratio of 140 ppt was determined by best fit comparison.*

*To provide some additional information on the choice of NO contamination levels, the following sentence has been added in Section 3.2.5: "The 140 ppt NO is based on contamination found in the $SO_2$ titration gas mixture during the OH reactivity comparison campaign (Fuchs et al., 2017) and the 20 ppt NO is based on the detection limit of the chemiluminescence analyser for NO concentration measurements at MOHp. The levels of concentration were also confirmed by direct NO measurements of the gas mixtures."*

P11 L5: Please double-check the uncertainty values stated for the scaling rate. How were they calculated? The 1σ determination using the IUPAC rate constant for OH+propane seems very small. The IUPAC compilation indicates that logk for OH+propane is ±0.08 at 298K, which I think translates into a 1 σ uncertainty of approximately 10%. How can the uncertainty on the scaling rate be lower than the uncertainty on the rate constant?

*Response: We double-checked the calculation, and indeed the underlying IUPAC uncertainty estimates were erroneous. They have been corrected for both OH + propane and OH + CO reactions. IUPAC provides estimates of expanded uncertainties at 95 % confidence level, and on page 5, as well as in the supplementary material (S2), 1 σ uncertainties (~simple uncertainty at 68 % confidence level) are quoted. The uncertainty in the scaling rate for propane has now been changed from 2 % to 5 %, and for CO has now changed from 15 % to 11 %.*

*The conclusions in Section 3.2.6 and the supplementary material S2 are not affected by this correction.*

P11 section 3.3: This section compares scaling rate values derived from flow tube calibrations to values inferred from atmospheric chamber experiments. This provides a nice validation of the flow

tube calibration method. The authors should also discuss the comparison of kw values derived from these different experiments? Were the kw values consistent within uncertainty?

*Response: The calculated $k_w$ values have been added to Table 2 and the comparison of $k_w$ is discussed in an additional paragraph at the end of Section 3.3 as follows:*

*"Comparing the wall loss rates from the 6[th] and 7[th] of April 2016 (i.e. external flow tube vs chamber), the $k_w$ values agree within one standard deviation, indicating the validity of the calibration approach using the external flow tube. However, as for the scaling rate, not all individual experiments at the SAPHIR chamber produced $k_w$ values that agree within one standard deviation, and the maximum difference in mean kw values is 1.2 $s^{-1}$.  Similarly, the individual $k_w$ measurements made at MOHp over the course of 6 months do not all agree within one standard deviation (Table 2), but the mean statistics of the SAPHIR chamber campaign and the MOHp long-term wall loss rate do agree within 1 σ."*

P12 L14-15: How is the upper limit for OH reactivity measurements quantified? Is the calculation based on a 3σ detection limit of OH in zone 2?

*Response: The upper limit is indeed based on the limit of detection of OH in zone 2. The quoted LOD of 5 x $10^5$ molecules $cm^{-3}$ for OH in Table 1 was confirmed by Berresheim et al. (2000) and is larger than the 2σ scatter of signal rate measurements, thus constitutes a conservative value. Therefore the quoted LOD has been used to determine the upper measurement limit for OH reactivity. In the modelled case where no NO is present in the system, the LOD for OH in zone 2 is reached around 40 $s^{-1}$, as evident in Figure 7b. As mentioned in the manuscript, the exact value of the upper limit of OH reactivity by this criterion (zone 2, [OH] LOD) will be variable as "the upper measurement limit is (…) dependent on the initially produced $[OH]_{UV}$ concentration".  Therefore it is important to note that the quantification here (i.e. 40 $s^{-1}$) is not an immutable number, and is based on model results with typical conditions ($[H_2O]$, T, p etc.) at MOHp. It is also approximately confirmed by calibration experiments with propane as OH reactant where CIMS OH reactivity shows underestimation above ca. 40 $s^{-1}$.*

*To clarify, the first sentence of Section 3.4 has been extended to quote the LOD criterion used: "The upper measurement limit for the described CIMS system is largely determined by the detectable OH concentration in titration zone 2 (i.e. the 2 σ [OH] LOD of 5 x $10^5$ molecules $cm^{-3}$, 5 min integration time, Table 1)."*

P12 L26: Sup. mat. S1 is cited for more details about the tagged mechanism. However there is no information about the tagging feature in S1.

*Response: The reactions where the recycled OH is tagged have been added to supplementary material S1:*

*"The following reactions are added to those in Table S1 when tagging for recycled OH (OHrec) is used and $H_2SO_4$ production pathways are separated:*

*OHrec + CO          =  H + $CO_2$*

| | | |
|---|---|---|
| OHrec + CO | = | HOCO |
| OHrec + $NO_2$ | = | $HNO_3$ |
| $HO_2$ + NO | = | OHrec + $NO_2$ |
| OHrec + NO | = | HONO |
| OHrec + $HO_2$ | = | $H_2O + O_2$ |
| OHrec + $C_3H_8$ | = | $IC_3H_7O_2$ |
| OHrec + $C_3H_8$ | = | $NC_3H_7O_2$ |
| OHrec | = | $N_2$ |
| | | |
| $SO_2$ + OHrec | = | $HSO_3$rec |
| $HSO_3rec + O_2$ | = | $SO_3OHrec + HO_2$ |
| $SO_3OHrec + H_2O + H_2O$ | = | $H_2SO_4OHrec + H_2O$ |
| | | |
| $SO_2 + HO_2$ | = | $SO_3HO_2$ + OHrec |
| $SO_3HO_2 + H_2O + H_2O$ | = | $H_2SO_4HO_2 + H_2O$ |
| | | |
| $SO_2 + IC_3H_7O_2$ | = | $SO_3RO_2 + IC_3H_7O$ |
| $SO_2 + NC_3H_7O_2$ | = | $SO_3RO_2 + NC_3H_7O$ |
| $SO_3RO_2 + H_2O + H_2O$ | = | $H_2SO_4RO_2 + H_2O$" |

P14 Eqs. 7-10: How was this complex empirical function determined? Can physical meanings be associated to some of the terms?

*Response: As Figure 8 illustrates, the response to the presence of NO in the system is non-linear for both increasing NO concentrations and OH reactivity, so non-linear functions were explored. As stated in the text, the empirical function was found by looking at the CIMS response for one given NO concentration one at a time: "By exploring CIMS OH reactivity measurements for each NO concentration (Fig. 9a), it was found that the correction could be empirically well described by an exponential equation of the form(…)". The fit parameters were determined by a Python (SciPy) curve fitting routine (i.e. finding a mathematical best fit solution) using all data (all NO concentrations and OH reactivity) and no attempt was made to create terms or parameters representative of physical processes. This is also already stated in the manuscript: "The fit parameters do not directly represent any physical or chemical processes or our understanding of the HOx recycling chemistry."*

*A different approach to describe the CIMS response was attempted by way of finding a solution to a simplified differential equation that described the expected chemical recycling reactions, radical production and loss processes (not shown or discussed in the manuscript). In that approach some approximate assumptions had to be made for which experimental data were lacking (such as e.g. $HO_2$ wall loss) and which could not be refined without further extensive experimental tests. As a consequence, the predictive capability of the more deterministic approach did not exceed that of the empirical eight-parameter exponential function. Finding a correction function with physically meaningful variables as input is current work in progress.*

P16 L22: A scatter of 10-20% is reported. Under which conditions was it observed (nature of trace gases, OH reactivity level, etc.). How many σ is it?

*Response: The full results from the OH reactivity comparison campaign at the SAPHIR chamber are described in detail in Fuchs et al. (2017); here a brief summary was added in the manuscript: "The scatter was greatest for the experiment of isoprene, MVK and MACR and the "urban mix" experiment with o-xylene, toluene, 1-pentene and was largest for OH reactivity above 15 $s^{-1}$. Since one standard deviation is at the order of 7 %, the reported scatter of up to 20 % is less than 3 σ of the measurements."*

P17 L7-9: Please be quantitative. How accurate is "high accuracy" and "lower accuracy"?

*Response: Mean deviation from reference OH reactivity has been included and the sentence now reads:*

*"The comparison including OH reactivity up to 40 $s^{-1}$ has shown that CIMS measurements were of high accuracy for certain chemical conditions (experiments with CO, pentane, monoterpenes, sesquiterpenes) where CIMS measurements and the OH reactivity reference agreed within uncertainty (mean deviation less than 13 %). For other conditions (isoprene, MVK, MACR mixture and urban mixture of o-xylene, toluene, 1-pentene) lower accuracy was observed, with a mean deviation to the reference of 27 %."*

Table 1: How many σ did the authors used to calculate the limits of detection? What are the precisions on Δt and kw values?

*Response: The limit of detection for OH concentration was determined by Berresheim et al. (2000) who wrote "From the 2σ scatter of signal count rate measurements at nighttime, the corresponding detection limits for 5 min signal integration were calculated to be 2 x $10^5$ molecules $cm^{-3}$ for OH (…). However, for the present time we conservatively estimate the OH detection limit to be 5 x $10^5$ molecules $cm^{-3}$". This conservative estimate has been found to be robust and stable and thus has been quoted here.*

*The limit of detection for kOH is a 1 σ value.*

*The precision on Δt is ± 5 %.*

*The precision on $k_w$ is ± 8 %.*

*All values/references have now been in included in Table 1.*

*Table 1: MOHp CIMS kOH measurement and system properties.*

| CIMS system property | value and measure |
|---|---|
| [OH] limit of detection | [a]$5 \times 10^5$ molecules cm$^{-3}$ |
| kOH limit of detection | [b]$0.5$ s$^{-1}$ |
| kOH upper measurement limit | $40$ s$^{-1}$ |
| typical $[OH]_0$ and $[HO_2]_0$ produced in UV zone | $10^8$ molecules cm$^{-3}$ |
| kOH temporal resolution | $60 - 300$ s |
| typical reaction time $\Delta t = (t_2-t_1)$ | [b]$0.1$ s ± 5 % |
| typical OH wall loss rate $k_w$ | [b]$10$ s$^{-1}$ ± 8 % |
| kOH measurement accuracy | [b]$1$ s$^{-1}$ (kOH < 30s$^{-1}$) |
| | [b]$2$ s$^{-1}$ (kOH 30-40 s$^{-1}$) |
| kOH measurement precision | [b]$0.14$ s$^{-1}$ |
| known interference | nitric oxide (NO) |

[a] *based on > 2 σ signal count rates, Berresheim et al. (2000).*

[b] *1 σ*

Fig. 14: It would be useful to add a time series for NO since it can disturb the measurements. Could the measurement underestimation due to the presence of NO also be quantified and shown as another time series?

*Response: Figure 14 has been adapted and another panel added. Since NO mixing ratios and the measurement underestimation are related to each other by a factor (0.7) they are referenced accordingly on the two y-axes of the panel. The left hand axis shows the measurement underestimation in s$^{-1}$ and the right hand axis shows the NO mixing ratio (ppb) at MOHp. The figure legend has been extended to include the added information.*

[Figure]

*"Figure 14: Observations at MOHp for the year 2015. (a) OH Reactivity measurements by CIMS, with 3 hour and daily means in blue and orange respectively. Filtered for recorded instrumental and laboratory interferences and seven outliers not coinciding with short term CO and $NO_2$ concentration peaks. 15541 valid measurement points. (b) Measured NO mixing ratios (right y-scale) and estimated CIMS OH reactivity measurement underestimation as a result of NO presence in ambient air (left y-scale)."*

Technical corrections

P1 L27: ". . .termed also total OH loss rate. . ." should read ". . .also termed total OH loss rate. . ."

*Response: Changed.*

P4 L20-23: This sentence should be rephrased. It's not clear what is meant by ". . . is the final OH concentration without UV light subtracted from the final OH concentration with the UV lamp on."

*Response: Some clarification is provided as to which individual OH measurements make up the OH reactivity measurement, and the sentence has been rephrased and extended:*

*"For the OH reactivity measurement in ambient air, OH concentrations are measured in titration zone 1 and 2 (Fig. 1), i.e. $[OH]_{T1}$ and $[OH]_{T2}$. In each titration zone, OH is both measured when the UV lamp is on and OH is produced in front of the inlet via R1 and R2, i.e. the same as during OH concentration calibration ($[OH]_{UV}$) as well as when the UV lamp is off, i.e. the same as during a normal OH concentration measurement, i.e. $[OH]_{ambient}$. The OH measurement used for OH reactivity is then the difference between the two measurements: $[OH]_{T\_1,2}$ = $[OH]_{UV\_1,2}$ – $[OH]_{ambient\_1,2}$. This way any*

*contribution from ambient OH on the OH reactivity measurement, typically at the order of a few percent, is removed."*

P4 Eq. 1: Missing negative sign in the exponential factor. kw should also be added in this equation for consistency with Eqs. 2-4

*Response: Corrected. Equation 1 is now:"* $[OH](t) = [OH]_0 \times e^{-(kOH + k_w) \times t}$ *with* $kOH = \sum k_{Xi+OH} \times [X_i]$ *and $X_i$ for the OH reactants and $k_w$ as the measurement system's wall loss rate. "*

P4 L32: Delete the word "lawful"

*Response: Deleted.*

P6 L7: ". . . within the flow tube from point of the inlet to . . ." should read ". . . within the flow tube from the inlet tip to . . ."

*Response: Changed.*

P6 L18-19: "The scaling rate, i.e. the times of SO$_2$/propane injection. . .". Shouldn't it read "The scaling rate, the inverse of the reaction time between T1 and T2. . ."

*Response: The referee is of course correct that the scaling rate is the inverse of the reaction time between T1 and T2. In the model the SO$_2$ and propane injections are triggered at the time steps required to achieve this specific reaction time between T1 and T2. This has now been clarified and the sentence modified to read: "The scaling rate and OH wall loss rate are prescribed based on experimental laboratory values. The scaling rate, as the inverse of the reaction time between T1 and T2, determines the SO$_2$ and propane injection in the model as the injections are triggered at the required times steps to achieve the prescribed reaction time."*

P7 L3-6: "volume flow" should read "volumetric flow rate". Several instances.

*Response: Changed for the instances found in this Section 3.1.*

P7 L12: "Additionally" should read "Additional"

*Response: Changed.*

P9 L26: "synthetic air has been measured for a range of VOC and OH reactivity calculated" should read "synthetic air has been screened for a range of VOC and OH reactivity calculated from a quantification of these contaminants"

*Response: Changed as suggested.*

P13 L18: Replace "annulled" by "cancelled"

*Response: Changed.*

P19 L24: "Calculations and propagation of uncertainties was performed. . ." should read "Calculations and propagation of uncertainties were performed. . ."

*Response: Corrected.*

P19 L27: Double-check the 5% value reported for LN([OH]T1/[OH]T2). It's 7.1% in section 3.2.1.

*Response: The value of 7.1 % is the correct one and the value on page 19 was rectified. For the illustrative example uncertainties were adjusted accordingly using the correct uncertainty value of 7.1 % for ln([OH]$_{T1}$/[OH]$_{T2}$). This resulted in minor changes of the values but does not affect the conclusion. Consequently, the NO concentration above the total measurement error is increased has been up-corrected to 370 ppt, i.e. the value changed from 330 to 370 ppt in the text:*

*"In this example the positive systematic error from the NO interference (defined in Section 3.5 as 0.8 s$^{-1}$ / ppb NO) would only be increasing the total measurement error for NO concentrations above 370 ppt."*

P19 L31-32. Some units are missing.

*Response: Unit for scaling rate (s$^{-1}$) included. The term ln([OH]$_{T1}$/[OH]$_{T2}$) has not got any units as it is the ratio between two concentration measurements.*

(End Referee#3)